# Amplified warming of seasonal cold extremes relative to the mean in the Northern Hemisphere extratropics

Mia H. Gross[a*], Markus G. Donat[a,b], Lisa V. Alexander[a], Steven C. Sherwood[a]

[a] Climate Change Research Centre and ARC Centre of Excellence for Climate Extremes, UNSW Sydney, Australia

[b] Barcelona Supercomputing Center, Barcelona, Spain

*Correspondence to*: Mia H. Gross (mia.gross89@gmail.com)

ORCID for M. H. Gross: 0000-0003-1589-6033

**Abstract**

Cold extremes are anticipated to warm at a faster rate than both hot extremes and average temperatures for much of the Northern Hemisphere. Anomalously warm cold extremes can affect numerous sectors, including human health, tourism and various ecosystems that are sensitive to cold temperatures. Using a selection of Global Climate Models, this paper explores the accelerated warming of seasonal cold extremes relative to seasonal mean temperatures in the Northern Hemisphere extratropics. The potential driving physical mechanisms are investigated by assessing conditions on or prior to the day when the cold extreme occurs to understand how the different environmental fields are related. During winter, North America, Europe and much of Eurasia show amplified warming of cold extremes projected for the late 21st century, compared to the mid-20th century. This is shown to be largely driven by reductions in cold air temperature advection, suggested as a likely consequence of Arctic amplification. In spring and autumn, cold extremes are expected to warm faster than average temperatures for most of the Northern Hemisphere mid- to high-latitudes, particularly Alaska, northern Canada and northern Eurasia. In the shoulder seasons, projected decreases in snow cover and associated reductions in surface albedo are suggested as the largest contributor affecting the accelerated rates of warming in cold extremes. The key findings of this study improve our understanding of the environmental conditions that contribute to the accelerated warming of cold extremes relative to mean temperatures.

**Keywords:** daily temperature, CMIP5, HadGHCND, snow cover, surface albedo, temperature advection

## 1. Introduction

Daily temperature extremes are expected to continue to warm, along with increases in mean temperatures, as a consequence of increasing greenhouse gases in the atmosphere. The rates of warming of extremes and mean temperatures are, however, not uniform, and differ depending on the season and region. Disproportionate rates of warming for different parts of the temperature distribution imply a change in the shape of the distribution. This is significant because it effects the probability and frequency of extreme events (Mearns et al., 1984), which can cause widespread impacts on both human and natural ecosystems, more so than changes in the mean temperature alone (Intergovernmental Panel on Climate Change (IPCC), 2012).

Both observational data and climate model simulations suggest that cold extremes are warming faster than warm extremes for much of the globe (e.g. Kharin and Zwiers, 2005; Donat and Alexander, 2012; Donat et al., 2013). Studies have also shown that in recent decades, cold extremes have been warming at a faster rate than local mean temperatures for some regions in the Northern Hemisphere (Brown et al., 2008; Gross et al., 2018). The amplified warming of cold extremes in these regions, relative to both the mean temperature and warm extremes, is indicative of decreasing variability during boreal winter (Screen, 2014; Ylhäisi and Räisänen, 2014; Schneider et al., 2015; Rhines et al., 2017). Climate model projections suggest this decrease in variability due to the accelerated warming of the coldest days will continue (Holmes et al., 2016), with cold extremes in some regions in the mid- to high-latitudes projected to increase over 5°C more than mean temperatures by the late 21$^{st}$ century (Gross et al., 2019). These disproportionate rates of warming suggest that changes in cold extremes are driven by mechanisms other than increases in local mean temperatures alone. A better understanding of the physical drivers related to the projected rates of the amplified warming of cold extremes is therefore crucial for assessing the probability and potential impacts of future changes in cold extremes.

The physical mechanisms driving the accelerated warming rates of cold extremes differ both regionally and seasonally. For land regions in the Northern Hemisphere mid- to high-latitudes, the warming of cold extremes and the associated decreases in temperature variability during winter months are consistent with reductions in advection of cold air that is a consequence of Arctic amplification (Screen, 2014; Schneider et al., 2015; Holmes et al., 2016; Rhines et al., 2017; Kanno et al., 2019). Arctic amplification, a phenomenon describing the enhanced warming of the Arctic relative to lower latitudes (Serreze and Francis, 2006), has been suggested as one of the dominant causes of the observed and projected reductions in the severity of extremely cold days during winter in the Northern Hemisphere extratropics (Screen, 2014; Schneider et al., 2015; Holmes et al., 2016; Rhines et al., 2017; Screen et al., 2018). This effect on cold extremes from Arctic amplification is shown to be a consequence of northerly winds from the Arctic bringing warmer than usual air to more southerly regions on the coldest days and reducing sub-seasonal temperature variability (Screen, 2014; Screen et al., 2014; Holmes et al., 2016). The loss of cold air has also accelerated in recent decades, with extremely cold air warming faster than moderately cold air (Kanno et al., 2019). Though it seems relatively clear that changes in temperature advection are linked with decreases in temperature variability in many mid- to high-latitude Northern Hemisphere regions, there is still uncertainty as to its role in driving the amplified warming of seasonal cold extremes relative to the corresponding seasonal mean. It is more likely that multiple factors are influencing the differences in seasonal and regional warming rates.

Aside from changes in atmospheric circulation patterns and thermal advection that may be altering cold extremes, variations in surface fluxes affecting the overall surface energy budget have strong links with surface temperatures and extremes. In particular, changes in snow cover play an important role in altering surface temperature in Northern Hemisphere regions that experience snowfall (e.g. Cohen and Rind, 1991; Mote, 2008; Diro et al., 2018). The high reflectivity and thermal emissivity of snow, compared to other natural surfaces, increases the surface albedo, lowers the absorbed shortwave radiation at the surface, and increases shortwave radiation reflected at the surface (Cohen and Rind, 1991). The effect of snow cover on surface temperature is greatest during spring when snow melt is at its highest, leading to increases in latent heat at the surface (Cohen and Rind, 1991; Dutra et al., 2011; Xu and Dirmeyer, 2011; Qu and Hall, 2014; Diro et al., 2018). Further, the surface albedo feedback stemming from snow cover is strongest during spring because insolation is low during winter months when snow accumulation is at its highest (Qu and Hall, 2014; Diro et al., 2018). The snow-temperature relationship is also affected by the snowpack, due to melting snow and consequent increases in latent heat, and vegetation cover, which acts to limit the role of snow cover and snow melt (Chapin III et al., 2005; Mote, 2008).

Climate model simulations have shown differences in the regions with the strongest snow-temperature relationship, with
some studies looking at North America finding the strongest links between temperature and snow cover over parts of eastern
North America (e.g. Xu and Dirmeyer, 2011), and others suggesting northwestern U.S. and southern Canada (e.g. Dutra et
al., 2011). Uncertainties related to biases within climate models are often related to the land cover parameterizations within
the models, such as how the models represent the masking effect of vegetation on snow cover (Loranty et al., 2014; Qu and
Hall, 2014) and how snow depth is treated within climate models (Mudryk et al., 2017). Evaluating the differences and
similarities between climate model simulations of snow cover, surface albedo and their influences may help to understand
sensitivities and increase confidence in future projections of warming.
This paper is structured by first evaluating a selection of Coupled Model Intercomparison Project phase 5 (CMIP5) climate
models (Taylor et al., 2012) against an observational dataset in terms of their ability to capture recent warming rates of
seasonal cold extremes relative to corresponding mean temperatures. This is followed by discussing predicted future changes
in the suite of climate models used. Next, the possible physical mechanisms driving the amplified warming of cold extremes
relative to seasonal means are explored. The investigated variables are chosen based on evidence that has been suggested by
prior studies, as previously discussed. We follow an approach similar to Donat et al. (2017), assessing conditions on the day
on which the cold extreme occurs, or conditions during the days directly prior to the day of the extreme.
**2. Data and Methods**
**2.1 Observational and CMIP5 data**
We use the Hadley Centre Global Historical Climatology Network-Daily (HadGHCND) dataset (Caesar et al., 2006) to
evaluate climate model simulations for the period 1950–2014. HadGHCND is a land only, daily gridded dataset of daily
maximum and minimum temperatures from ground stations, for which daily mean temperatures are calculated by taking the
average of each daily maximum and minimum temperature value for each grid cell.
The HadGHCND data are used to evaluate six individual CMIP5 models (see Table 1), which were selected based on their
data availability for all of the daily climate variables being investigated. While we only show a single simulation from each
model (r1i1p1), multiple ensemble runs were analysed (where available) to determine model robustness and assess internal
climate variability within the models. Results of multiple ensemble runs were found to be highly correlated in both spatial
pattern and magnitude of simulated changes (see Fig. S1), indicating that sensitivity of the results to internal variability
within the models is small. Historical model simulations (1950–2005) are merged with Representative Concentration
Pathway 8.5 (RCP8.5) simulations (2006–2099) to assess changes between the mid-20th century and early 21st century
(1950–2014), as well as between the mid-20th century and late 21st century (1950–2099). For analysis of recent decades, a
bilinear remapping technique is used to re-grid all models to the grid cell size of HadGHCND, that is, 2.5° latitude x 3.75°
longitude, and masked to only cover land regions where sufficient observational data are available. We define 'sufficient' as
being grid cells with at least 80% of daily data available over 1950-2014, as well as at least 50% of data available for the
first and last ten years of observational data. For analysis of future projected changes, all models are re-gridded to a common
grid size of 2.5° latitude x 2.5° longitude to enable inter-model comparison and analysis of the multi-model mean.
**2.2 Methods**

For each model simulation as well as HadGHCND, daily temperature anomalies are calculated relative to a mean annual

cycle of daily mean temperatures based on the entire period of analysis (1950–2014 for analysis of recent changes, and

1950–2099 for analysis of future changes). The data are then split into seasons, boreal winter (December to February – DJF),

spring (March to May – MAM) and autumn (September to November – SON), and all analyses are only applied to Northern

Hemisphere land areas north of 30°N. Boreal summer is not included in the analysis as it was previously found to have only

small changes in cold extremes relative to the mean that were less robust across a suite of CMIP5 models (Gross et al.,

2019).

For each grid cell in each dataset, the seasonal minima of daily temperature anomalies are calculated annually for 1950–2014

and 1950–2099 separately, accounting for the differences in base period selection. The seasonal minima are then averaged

over two periods, 1950–1981 and 1982–2014 for analysis of recent changes, and 1950–1979 and 2070–2099 for analysis of

future changes, to calculate changes in the anomalously coldest days. Changes in seasonal mean temperature is similarly

computed from daily mean temperature data. The difference between changes in the seasonal minima and changes in the

seasonal mean is then calculated, hereafter referred to as 'excess changes'. 'Recent excess changes' refer to excess changes

between the mid-20th century and early 21st century, while 'future excess changes' refer to excess changes between the mid-

20th century and late 21st century. Local significance of future excess changes is assessed by a Kolmogorov-Smirnov test

(KS-test) at the 5% level.

To investigate the possible drivers of the amplified warming of seasonal cold extremes relative to the mean in the mid- to

high-latitude Northern Hemisphere regions, we assess several variables available at the daily time-scale in the selected

CMIP5 models. This includes snow cover (CMIP5 variable name *snc*), snow amount (*snw*) and upwelling and downwelling

longwave and shortwave radiation fluxes at the surface (*rlus*, *rlds*, *rsus*, *rsds*). We also assess surface albedo, calculated as

the ratio of upwelling shortwave radiation and downwelling shortwave radiation, and horizontal temperature advection,

which is derived for each model using the equation:

$$\frac{\partial T}{\partial t} = -(u\frac{\partial T}{\partial y} + v\frac{\partial T}{\partial y})$$

where $\partial T/\partial t$ is the horizontal temperature advection in °C/s, u and v are the zonal and meridional wind components (*uas* and

*vas*, respectively), and $\partial T/\partial x$ and $\partial T/\partial y$ are the temperature gradients in the zonal and meridional direction. We refer to

advection of cold air temperature hereafter as 'negative temperature advection'. For surface albedo, there are some instances

in high-latitude regions where values are unrealistically large, as a result of low incoming shortwave radiation values that

affect the calculation of surface albedo. In any instance where surface albedo values are outside of the physically reasonable

0 to 1 range, values are set to missing. Several other daily variables were also assessed, such as surface heat fluxes and cloud

cover, but were found of low relevance as potential drivers of cold extremes in the seasons and regions being examined.

The analysis of the physical mechanisms related to the amplified warming of cold extremes is limited to future changes,

where the signal is stronger than for recent changes, and therefore shows a more robust identification of relationships. For

each of the variables assessed, except temperature advection, data are evaluated on the specific day when the seasonal

minima occurs. For temperature advection, a three-day average prior to the day the cold extreme occurs is used. This is

because it is likely that larger changes in circulation would have more of an influence on temperature in the days leading up

to the event, rather than the day of the event. A three-day average leading up to the day of the cold event was also assessed

for snow cover and albedo, but results showed no clear difference compared to using values on the exact day of the event.

Excess changes are also calculated for each variable in much the same way as excess temperatures, that is, taking the

difference between the value of the variable on the days of the cold extreme (or three-day average prior to the event for

temperature advection) and the seasonal mean of the variable. This essentially removes the mean from the analysis and
shows regions that experience increases or decreases in conditions related entirely to the days on which the cold extremes
occur.

Results of the physical relationships are presented in two ways: maps of the variables as is shown for excess changes in
temperature (to infer on the similarity of spatial patterns), and scatter plots of correlations of future excess changes in cold
extremes with either snow cover or albedo. The former is included in supplementary material while the latter are included
within the main body of the manuscript. For the scatter plots, seasonal 'excess' values for the two time periods used for the
future analysis are calculated as the difference between the variable value on the day the seasonal minima occurs and the
seasonal mean of the respective variable. For simplicity, we use the term 'actual changes' to refer to changes in the actual
values of the different variables on the days the cold extremes occur (or the three-day average prior to this day for negative
temperature advection). Weighted area-averages of the annual excess values are then calculated for all grid boxes within a
selected region that adhere to a specified condition that only includes grid cells with a statistically significant future excess
change exceeding 1°C. Two regions are assessed for all models, one covering North America (30°N to 70°N, 168°W to
52°W) and the other covering much of northern Eurasia (47°N to 75°N, 10°E to 135°E) (see Fig. S2). Regressions are
calculated using total least squares regression, with correlation coefficients computed using Spearman's rank correlation.

**3. Results**

**3.1 Recent changes in cold extremes relative to the mean**

Historical excess changes in seasonal cold extremes relative to corresponding mean temperatures are shown for HadGHCND
and the six-member multi-model mean for boreal winter, spring and autumn (Fig. 1). Maps of individual models are included
as supplementary material (Figs. S3–S5). Positive values indicate regions where cold extremes have warmed more than the
mean, while negative values indicate regions where cold extremes have warmed less than the mean. Stippling indicates grid
cells where both five out of six models agree on the sign of excess change and where the multi-model mean agrees in sign
with HadGHCND.

During winter (Fig. 1a), HadGHCND shows that cold extremes have warmed more than 1°C faster than the mean for parts of
northern and eastern Canada, western U.S., eastern Siberia and parts of northern and central Eurasia. The mean has warmed
more than cold extremes in parts of western Alaska and north-western Russia. In spring (Fig. 1b), much of North America
shows strong positive excess changes, excluding eastern Canada which shows some negative excess changes around -1°C.
Positive excess changes over 1°C in spring are also shown for Nordic countries and eastern Russia, while central-northern
Russia shows some areas of negative excess changes. In autumn (Fig. 1c), much of eastern North America, excluding eastern
Canada, shows strong positive excess changes sometimes exceeding 1°C. This is similarly shown for parts of Siberia, the
Nordic countries and Eurasia, while negative excess changes are evident over western Russia and parts of Europe and the
Mediterranean region.

The CMIP5 multi-model mean shows a smoother spatial pattern overall compared with HadGHCND, with agreement
between observations and the models themselves mostly in regions where the strongest positive excess changes are seen.
Though underestimating excess changes in HadGHCND, there is strong agreement in both the sign of the individual CMIP5
models and with HadGHCND that cold extremes have warmed more than the mean for parts of northern and eastern Canada

and northern and central Eurasia during winter (Fig. 1d). While the multi-model mean shows small negative excess changes for eastern Siberia, individual models disagree in sign, with MPI-ESM-LR showing positive excess changes in the region similar in magnitude to HadGHCND (Fig. S3). During spring (Fig. 1e), there is strong model agreement and similarities with HadGHCND for much of central Eurasia, the Nordic countries and northern North America, with cold extremes warming between 0.4°C and 0.8°C more than the mean during recent decades. Some of the individual models show stronger changes than others in these regions, such as INM-CM4 and CanESM2 (Fig. S4). Autumn shows a similar pattern to spring in the multi-model mean, with slightly more model agreement over a larger part of northern Eurasia (Fig. 1f). The models agree on the positive excess changes shown in HadGHCND over north-eastern North America, southern Greenland, the Nordic countries, Siberia and central Eurasia, though they do not capture the negative excess change over western Russia and Europe. This negative excess change, however, is shown to a lesser degree in CanESM2 and MPI-ESM-MR (Fig. S5). Though the multi-model mean underestimates the positive excess changes in HadGHCND, excess changes simulated in the individual models vary, with some resembling the magnitude shown in HadGHCND greater than others.

The shoulder seasons generally show widespread positive excess changes in the multi-model mean and individual models, with agreement between models as well as with HadGHCND over much of Eurasia and northern North America. Winter also shows strong agreement over some of these regions. Across all seasons shown, the same general pattern of excess changes in cold extremes is clear, with the most prominent positive excess changes in recent decades occurring in the northern continental interiors. This motives us to assess how cold extremes might change in the future relative to mean temperatures in the selected six climate models over the Northern Hemisphere extratropics.

**3.2 Projected excess changes in cold extremes**

Projections of excess changes in cold extremes comparing the mid-20$^{th}$ century with the late 21$^{st}$ century are shown for boreal winter, spring and autumn using the six-member multi-model mean (Fig. 2). Future excess changes in the individual models are included as supplementary material (Figs. S6–S8).

Cold extremes are projected to warm significantly more than mean temperatures across much of the Northern Hemisphere extratropics. During winter, the amplification of cold extremes relative to the mean is strongest in Alaska, eastern and western Canada, Nordic countries and north-western Eurasia, with positive excess changes exceeding 5°C in some of these locations. Similar to the historical excess changes for winter, CanESM2 projects the largest positive excess changes, however, all six of the models show positive excess changes of at least 2°C in these regions (Fig. S6). There is some variation over northern Russia and Siberia, with CNRM-CM5 showing significant negative excess changes around -1.5°C, opposed to relatively strong positive excess changes of around 3°C in CanESM2, MPI-ESM-LR and MPI-ESM-MR. The shoulder seasons also show a similar spatial pattern to historical excess changes, albeit at a greater magnitude, with positive excess changes exceeding 3°C projected for most of the Northern Hemisphere extratropics. For both spring and autumn, the largest excess changes are projected for northern North America and northern parts of Eurasia, in particular western Russia and the Nordic countries but also spreading further east in autumn into the eastern parts of Siberia. During spring, the models show some differences in southern parts of the U.S. and Eurasia, which mostly show non-statistically significant negative excess changes (Fig. S7). In autumn, which shows the largest and most widespread excess changes, there is robust model agreement that cold extremes are projected to warm in excess of 5°C more than the mean for much of northern North America and northern Eurasia. In the individual models, the excess changes in these regions range from around 3°C in INM-CM4 to over 6°C in CanESM2 (Fig. S8).

Future excess changes are robust and systematic, with strong model agreement that cold extremes are expected to warm

more than mean temperatures for many mid- to high-latitude regions in boreal winter, spring and autumn. As in the historical

excess changes, spring and autumn generally show a more widespread and systematic pattern of positive excess changes

over Eurasia, Canada and Alaska, however, the projected magnitude of amplified cold extremes relative to mean

temperatures exceeds 3°C for much of the Northern Hemisphere extratropics in seasons shown. To explore the possible

physical mechanisms driving the amplified warming of cold extremes, we focus on the regions that show the most robust

signals. The strongest excess changes across all the shown seasons are over northern Eurasia and northern North America.

This is relatively consistent with the largest recent excess changes occurring in the northern continental interiors in

observations and CMIP5 models, though is much more widespread and systematic in the projected patterns.

**3.3 Projected changes in advection of cold air prior to cold extremes**

Due to the evidence suggesting Arctic amplification, and consequent changes in thermal advection, as a main driver of

decreasing temperature variability in Northern Hemisphere regions (e.g. Screen and Simmonds, 2010; Screen, 2014;

Schneider et al., 2015; Holmes et al., 2016; Rhines et al., 2017), we first consider projections of changes in temperature

advection averaged over the three days prior to the cold event. Figure 3 shows future changes in actual and excess

temperature advection in the six-member CMIP5 multi-model mean, with stippling indicating grid cells where at least five

out of six models agree on the sign of change. As described in Sect. 2.2, actual changes refer to changes only in the days

prior to the day of the extreme, while excess changes show the difference between the days prior to the extreme and the

seasonal mean temperature advection. Results of individual models are included as supplementary material (Figs. S9–S10).

The most notable features occur for boreal winter, where the multi-model mean projects reductions in negative temperature

advection for much of North America as well as Eurasia, corresponding to reduced advection of colder air in these regions.

This is evident for changes in both actual and excess negative temperature advection, which suggests the changes are related

specially to the days directly prior to the day the cold extreme occurs, rather than to changes in average seasonal advection

of cold air. While there is strong agreement between models regarding this, the magnitude of the reduction in the advection

of cold air varies between individual models, where CanESM2 generally shows the largest reduction compared with the

other models (Figs. S9–S10). This is reflected in future excess changes in cold extremes, where CanESM2 is generally

warmer than the other models during winter (see Fig. S6). For much of North America, especially central and eastern U.S.

and south-western Canada, reductions in negative temperature advection of at least -2°C/s are projected for the late

$21^{st}$ century, with the same areas showing the largest projected positive excess changes in cold extremes. This is indicative of

reduced cold air temperature advection, related to both the day the cold extreme is projected to occur as well as changes in

the mean seasonal temperature advection, being a dominant driver of the amplified warming of cold extremes relative to the

mean during boreal winter. Similarly, the greatest decreases in negative temperature advection over the European continent

occur in the Nordic countries and Eurasia, which also show high magnitude excess warming in cold extremes during

winter. This same pattern is evident across all of the selected climate models (Figs. S9–S10). Though some reductions in

negative temperature advection are shown scattered over North America and Eurasia for spring and autumn, the spatial

pattern does not match with the seasonal future excess changes in cold extremes like it does for winter.

Based on these results, it is evident that a reduction in the advection of cold air is driving the projected excess changes in

cold extremes over much of North America and Eurasia during winter. Both shoulder seasons, however, show a less clear

signal with generally smaller changes in negative temperature advection, pointing to other mechanisms being a dominant

driver of the projected amplified warming of cold extremes in spring and autumn.

**3.4 Projected changes in snow cover and surface albedo associated with cold extremes**

Many of the grid cells showing significantly strong excess changes are located in regions that experience high seasonal snow cover. Snow cover and associated surface albedo feedbacks therefore play a major role in temperature variability in these regions, but it is not clear if this relationship extends to the amplified warming of cold extremes relative to local mean temperatures, and the seasonal influence remains uncertain. The subsequent results show scatter plots of excess changes in cold extremes and snow cover (Fig. 4) and surface albedo (Fig. 5). As outlined in Sect. 2.2, these changes are calculated for the exact day when the cold extreme occurs. Projections of changes in actual and excess snow cover and surface albedo for the days of the cold extreme are included as Supplementary Material (Figs. S11–S12 and S13–S14, respectively). For additional information on the snow-temperature relationship, future changes in snow amount are also included as Supplementary Material (see Figs. S15–S16).

For both regions, mostly significant negative correlations between snow cover and excess cold extremes are shown for all seasons, aside from excess snow cover in boreal winter. During winter in the North America region (Fig. 4a-f), all models show significant negative correlations of at least -0.74 for actual snow cover (Fig. 4a), however, all models except CSIRO-Mk3-6-0 show significant positive correlations for excess snow cover (Fig. 4b). From Figs. S11–S12, parts of North America, particularly southern Alaska, southern Canada and along the north-western coast of the U.S., show projected decreases in actual snow cover, but slight increases in excess snow cover. This suggests that the feedback between snow cover and the projected amplified warming of cold extremes is related to overall reductions in the seasonal mean snow cover during winter, rather than decreases in snow cover on the day the cold extreme occurs. Negative correlations in spring and autumn are generally stronger than they are for winter, in both actual and excess snow cover (Fig. 4c,e and Fig. 4d,f respectively), with the greatest overall correlations projected for autumn. Again, this is reflected in the maps, where actual snow cover is projected to decrease around 40% for much of Alaska and northern Canada during autumn, while decreasing somewhat less and slightly further south during spring (Fig. S11). Smaller decreases are projected for excess snow cover, compared with actual snow cover, excluding Alaska which shows mostly small increases during spring (Fig. S12), implying that projected decreases in the overall mean-state of snow cover are related to projected excess changes more than decreases in snow cover on the day of the extreme. Northern Eurasia (Fig. 4g-l) shows similar correlations to that of North America. The overall largest correlations between snow cover and excess cold extremes occur in autumn (Fig. 4k,l), with some models, for example, CanESM2 and CNRM-CM5, showing correlations as high as -0.91 (Fig. 4k). In these models, decreases in snow cover over 45% are shown for parts of western Russia and Scandinavia (Fig. S11). Correlations are slightly lower for spring (Fig. 4i), with the largest projected spring decreases in actual snow cover shown for the Nordic countries and central/eastern Europe, with no substantial changes in Siberia (Fig. S11). This is reflected in projected changes in actual snow amount (Fig. S15), with increases shown for regions that project no changes in snow cover. The lack of snow cover changes in the coldest climates, such as in Siberia, is likely due to the trade-off between increasing temperatures that shorten the snow season and increased moisture holding capacity which leads to greater snowfall in these regions (e.g. Krasting et al., 2013; Mankin and Diffenbaugh, 2015). Correlations with excess snow cover in spring (Fig. 4j) are substantially smaller for most models, compared with actual snow cover, with parts of northern Russia showing small increases in snow cover (Fig. S12) and snow amount (Fig. S16).

Decreases in snow cover imply that reductions in surface albedo are a likely factor contributing to the amplified warming of cold extremes relative to the mean. Correlations between surface albedo and excess cold extremes (Fig. 5) indeed show strong similarities with those of snow cover, with the largest negative correlations shown for boreal autumn for both North

America (Fig. 5a-f) and northern Eurasia (Fig. 5g-l). As shown for snow cover, the strongest overall projected decreases are shown for actual changes in surface albedo over Alaska, northern Canada and Eurasia during autumn months (Fig. S13). Differences in the magnitude and sign between actual surface albedo and excess surface albedo are also clear (Fig. S13–S14). Mostly positive correlations with excess surface albedo are shown for winter for both regions (Fig. 5b,h). During boreal winter in high-latitude regions, solar insolation is low, so it is expected that surface albedo is less of a factor in driving excess changes in cold extremes during the winter months.

There is a clear relationship between decreases in snow cover, associated lower albedo and the amplified warming of cold extremes for many regions in the Northern Hemisphere mid- to high-latitudes. While negative correlations are shown for actual snow cover and excess cold extremes during winter for both North America and Eurasia, projected decreases in actual snow cover, as shown in the maps in Fig. S11, are generally much smaller than they are for both shoulder seasons, especially autumn months which show the overall largest decreases and highest correlations with excess temperatures in cold extremes. Much of this relationship between snow cover, surface albedo and excess temperatures in cold extremes is a consequence of overall decreases in the mean-state of both snow cover and surface albedo, rather than decreases in snow cover specifically on the day in which the cold extreme occurs. This is consistent across the selection of CMIP5 models used in this study.

**3.5 Projected changes in the timing of anomalously cold days**

The amplified warming of cold extremes projected for much of the Northern Hemisphere mid- to high-latitudes is related to excess heat near the land surface that acts to decrease the severity of the anomalously coldest days of the season. During spring and autumn, much of this is likely a consequence of less snow cover and lower albedo, leading to increases in absorbed shortwave radiation at the surface and consequently amplifying the warming of cold extremes, creating a positive feedback within the system. In addition to these relationships, we also analysed an increase in net radiation on the days of the cold extremes in both shoulder seasons, with increases in incoming shortwave radiation being the largest contributor (not shown). These increases are, however, largely attributable to temporal shifts in the occurrence of the largest negative temperature anomalies in the shoulder seasons.

Figure 6 shows the projected change in the timing of the seasonal minimum of daily anomalies in the six-member multi-model mean (see Fig. S17 for individual model results). Positive values indicate grid cells where the coldest days are projected to occur later in the season, while negative values indicate grid cells where the coldest days are projected to occur earlier in the season. Changes in the anomalously coldest winter day are mostly small with little model agreement, except for far-eastern Canada where the coldest winter days are projected to occur between 8 and 16 days earlier in the season (Fig. 6a). The shoulder seasons both show more significant shifts in the timing of the anomalously coldest days. For much of the Northern Hemisphere mid- to high latitudes, excluding the most southerly parts, the Mediterranean region and parts of Greenland, the anomalously coldest days are projected to occur later in the season during spring (Fig. 6b). In some regions, such as central-western Europe and eastern Canada, the anomalously coldest spring days are projected to occur more than 20 days later in the late 21$^{st}$ century, compared to those simulated in the mid-20$^{th}$ century. Some models, such as CanESM2, project over a 30-day shift in the timing of spring cold extremes in these areas (Fig. S17). During autumn (Fig. 6c), the anomalously coldest days are projected to shift to earlier in the season for most high latitude regions in the Northern Hemisphere. For example, in the multi-model mean, the anomalously coldest days are projected to occur up to about 30 days earlier than they did in the mid-20$^{th}$ century in some regions in northern Canada and northern Eurasia. This change in the timing of anomalously cold days suggests an overall flattening of the seasonal cycle in these extratropical Northern Hemisphere regions. Coupled with the cold extremes warming at a faster rate than average temperatures, this suggests these

382 regions will generally experience a longer duration warm season and a shorter duration cold season, in confirmation with
383 previous studies (e.g. Dwyer et al., 2012; Chen et al., 2019).

**4. Discussion and conclusions**

Cold extremes are projected to warm more than seasonal average temperatures for much of the Northern Hemisphere mid- to
high-latitude regions, for all seasons except boreal summer. Though these projected changes differ slightly in magnitude and
spatial pattern depending on the CMIP5 model used, the most prominent excess changes are robust across the selection of
models. These changes are likely related to projected changes in horizontal temperature advection, snow cover and surface
albedo feedbacks. The season in which the excess changes in cold extremes occur largely dictates which physical
mechanisms are at play.

Decreases in snow cover and surface albedo are more associated with excess changes in cold extremes during spring and
autumn months. Due to low solar insolation in winter months, and subsequently only small effects from changes in
shortwave radiation and surface albedo, reductions in advection of cold air in the days leading up to the extreme event is the
dominant driver during boreal winter. This latter finding is likely a consequence of Arctic amplification and is in agreement
with previous studies linking the warming of cold days in winter months with warmer than usual air being brought from the
Arctic to lower latitudes (e.g. Screen, 2014; Schneider et al., 2015; Holmes et al., 2016; Rhines et al., 2017).

In contrast, Arctic warming and associated sea ice loss has been argued to result in more persistent severe cold air outbreaks
over continental regions in the mid-latitudes during boreal winter (e.g. Kodra et al., 2011; Cohen et al., 2014, 2018; Francis
and Vavrus, 2015; Zhang et al., 2016). However, atmospheric circulation is argued to play a more substantial role in
influencing cold winters compared with Arctic sea ice loss (Blackport et al., 2019). Recent cold snaps in the United States
and Eurasia, such as those observed in the boreal winter of 2012/2013, can largely be explained by a southward shift in the
jet stream and a weakening of the stratospheric polar vortex (Francis and Vavrus, 2015; Zhang et al., 2016; Cohen et al.,
2018; Kretschmer et al., 2018). Though some argue these events are likely transient and related to atmospheric decadal
variability (e.g. Barnes and Screen, 2015; Ayarzagüena and Screen, 2016; Sun et al., 2016), others suggest that severe cold
snaps in the Northern Hemisphere mid-latitudes might persist in response to continued Arctic warming (e.g. Kodra et al.,
2011; Francis and Vavrus, 2012, 2015; Cohen et al., 2014). While there is some disagreement between models and
observations in how they simulate the observed cold outbreaks (e.g. Cohen et al., 2013; Sun et al., 2016), there is robust
model agreement that mid-latitude cold extremes are projected to decrease in severity (Screen, 2014; Barnes and Screen,
2015; Screen et al., 2015a, b; Ayarzagüena and Screen, 2016). Some have also suggested that cold air outbreaks are expected
to decrease in duration and frequency (e.g. Screen et al., 2015a, b), however, this remains unclear and requires further work
(e.g. Ayarzagüena and Screen, 2016). Though the results in this study cannot infer anything regarding the frequency and
duration of cold spells, it is evident that cold extremes are projected to warm in excess of increasing mean temperatures over
much of North America and Eurasia during boreal winter by the end of the 21$^{st}$ century. Though lacking model agreement,
small negative excess changes are projected for parts of Eurasia, such as central-eastern Asia and northern parts of Siberia
(Fig. 2a). This is also evident in historical excess changes (Fig. 1a). This is consistent with the 'warm Arctic, cold Eurasia'
pattern relating to substantial sea ice concentration in the Barents-Kara seas and high-latitude blocking associated with a
positive phase of the North Atlantic Oscillation (Luo et al., 2019a; Luo et al., 2019b). A larger model ensemble would be
useful to further quantify whether this pattern is robustly projected for amplified cold extremes.

Arctic amplification and associated thermal advection is also suggested to be a particularly strong driver of the decreased severity of cold extremes in autumn months (e.g. Screen, 2014; Holmes et al., 2016). Even though some reductions are projected in the advection of cold air during autumn (Fig. 3c), reductions during winter are far greater with a much clearer link to projections in excess cold extremes. Projected changes in negative temperature advection during spring show a similar pattern to changes in autumn. While Arctic amplification and associated reductions in the advection of cold air may be having somewhat of an impact on the warming of cold extremes during the shoulder seasons, other physical mechanisms likely have a greater influence on changes in spring and autumn cold extremes.

For both shoulder seasons, 'hot spots' of amplified warming of cold extremes relative to the mean are shown for much of Alaska, Canada and northern Eurasia (Fig. 3b,c). During autumn, changes in snow cover show an exceptionally similar spatial pattern to excess changes in cold extremes for all models (see Fig. S11 and Fig. S5, respectively), with the largest excess changes in cold extremes matching regions showing the largest decreases in snow cover. Spatial similarities between snow cover and excess changes in cold extremes during spring are less obvious than they are for autumn, with slightly lower correlations, though the largest decreases in snow cover are still associated with significant excess changes in cold extremes. Previous work has suggested that spring has the strongest snow-temperature relationship, largely due to increases in latent heat from snowmelt (e.g. Dutra et al., 2011; Xu and Dirmeyer, 2011; Diro et al., 2018). Many of the regions showing the strongest relationship between projected snow cover and the projected amplification of warming cold extremes, such as north-western U.S., southern and northeast Canada and the Rocky Mountains, are in agreement with historical simulations of the snow-temperature association during winter and spring months (Dutra et al., 2011; Diro et al., 2018). While some high-latitude regions in northern Canada and northern Russia show projected increases in snow amount during spring (Figs. S15–S16), with the same regions and seasons showing no substantial changes in snow cover (Figs. S11–S12), correlations between snow cover and excess temperature in autumn are generally larger. This infers that even if springtime is associated with a stronger snow-temperature relationship, due to increases in snowmelt, decreases in snow cover have more of an influence on warming anomalously cold days in autumn months.

A change in surface albedo feedback, as a result of a change in snow cover, is more likely to influence cold days in early spring, compared to winter, due to snow accumulation and low insolation during winter months. While results presented here show projections of decreasing albedo for many regions in North America and Europe, autumn shows the largest decreases in surface albedo (see Fig. S13), which is closely related to the projected decreases in snow cover. We note that our calculation of surface albedo may not be capturing certain aspects that are important to snow affected areas. For example, the boreal forest is a region with extensive snow fall and dense vegetation cover, and the varying land parameterizations within the climate models may not necessarily be capturing the snow that is intercepted by trees in the canopy (Loranty et al., 2014; Thackeray et al., 2015). This then has important implications on surface albedo and therefore surface temperature. Biases in climate model simulations of snow-albedo feedbacks have been found over the boreal forest region, with significant underestimations compared with observations, especially during periods where snowmelt is high, such as in early March (Fletcher et al., 2012; Loranty et al., 2014; Qu and Hall, 2014; Thackeray et al., 2014, 2015). However, biases in the models are reduced over larger study regions (Thackeray et al., 2015), with area-averaging over large regions also likely to suppress any biases. Biases may also simply be a consequence of temperature, with cold biases having more snow, and warm biases leading to more snowmelt. The ability of climate models to capture snow-albedo feedbacks is also complicated by factors relating to snow type and the ageing of snow, which can also influence surface temperatures (Thackeray et al., 2015; Diro et al., 2018). Previous work has found that climate models tend to underestimate snow-albedo feedbacks compared with observations (Brutel-Vuilmet et al., 2013; Qu and Hall, 2014), which is potentially tied to models underestimating the sensitivity of snow cover to warming (Mudryk et al., 2017). Improving the ability of climate models to capture realistic

changes in snow cover and surface albedo would enable more accurate projections of future cold extremes. Biases in the representation of physical relationships may control the simulation of long-term changes in cold extremes. Given the availability of suitable observations of relevant land variables, an evaluation of the land-atmosphere relationships as outlined here may serve to develop process-based constraints to reduce the uncertainty in future projections, similar to previous approaches focussing on the processes driving hot extremes in summer (Donat et al., 2018).

While our findings are consistent with the theory that less snow cover and associated reductions in surface albedo lead to anomalously warmer temperatures on cold days, it is unclear whether these variables are driving the amplified warming of cold extremes, or vice versa. It is true, however, that the positive feedback between snow cover, surface albedo and surface temperature exacerbate the warming of cold extremes. It would be useful for future studies to run climate model simulations with and without snow cover prescribed to quantify the specific impact on simulated cold extremes, enabling more confident conclusions regarding snow cover and albedo as a driver of amplified warming of cold extremes.

Similar to albedo, radiative fluxes are strongly influenced by changes in the surface which affects the overall surface energy budget. For example, increased moisture load and associated enhanced downward longwave radiation have been shown to play an important role in Arctic amplification (Lee et al., 2017; Luo et al., 2017a). Decreases in snow cover which lead to lower albedo will result in increased absorption of incoming shortwave radiation for regions and seasons with enough solar insolation. While we did find some increases in incoming shortwave radiation on the days when the coldest anomalies occur, this is more a consequence of the timing in which the cold extremes occur. For high-latitude regions, the seasonal minimum temperature anomaly in spring is projected to occur later in the season, with the coldest autumn day projected to occur earlier in the season, suggesting an overall flattening of the seasonal cycle. Changes in the annual cycle of surface temperature have been detected before, with a shift to earlier seasons by 1.7 days from 1954 to 2007 over land in the extratropics (Stine et al., 2009). Recent methods used to detect changes in the annual cycle highlight the importance of using a changing, time-dependent amplitude to account for variability in anomalies (e.g. Deng et al., 2017; Deng et al., 2019). Changes in the seasonal cycle have previously been shown in CMIP3 and CMIP5 models as well, with colder temperatures occuring later in the season and warmer temperatures occuring earlier, reducing the amplitude of the seasonal cycle in high-latitude regions (Dwyer et al., 2012; Chen et al., 2019). These shifts are argued to be a consequence of anthropogenic climate change driving sea ice loss (Dwyer et al., 2012; Chen et al., 2019), but have also been linked with changes in the Northern Annular Mode (Stine et al., 2009; Luo et al., 2017b).

The projected anomalous coldest day during spring and autumn is also associated with less snow, albeit largely due to projected decreases in mean seasonal snow cover. Because this day is occurring closer to summer in both seasons, there will be greater snowmelt. This describes another positive feedback within the system, with snowmelt leading to increases in latent heat which in turn heats the surface. This highlights the fact that multiple factors within the surface energy budget are contributing to an overall greater heating at the surface, thus influencing the decrease in the severity of cold days relative to mean warming during spring and autumn months.

The amplified warming of seasonal cold extremes relative to seasonal mean temperature is projected for much of the Northern Hemisphere mid- to high-latitudes. The main findings of this paper show that the possible drivers of this amplified warming depend on the season. Reduced advection of cold air as a consequence of Arctic amplification is the dominant driver of projected amplified cold extremes during boreal winter. For autumn and spring, projected decreases in snow cover and lower surface albedo contribute to the projected accelerated warming of cold extremes. These findings are robust across the selection of CMIP5 models used in this study. While observational data were used to evaluate simulations of excess

temperature in recent decades, the possible drivers are only explored as future changes, with model agreement suggesting how robust the changes are. Further work in understanding the physical mechanisms driving cold extremes would benefit from further evaluation of observational data of snow cover, wind and surface radiation fluxes against model simulations used to predict future excess changes.

**Author contributions**

MHG has performed the analyses and drafted the manuscript, with contributions from all co-authors.

**Data availability**

HadGHCND data can be downloaded for free at http://www.metoffice.gov.uk/hadobs/hadghcnd/download.html. All CMIP5 data are available from the public CMIP5 archive and can be downloaded at https://esgf-node.llnl.gov/search/cmip5.

**Competing interests**

The authors declare that there are no conflicts of interest.

**Acknowledgments**

This study was supported by the Australian Research Council (ARC) Centre of Excellence for Climate Extremes (Grant CE170100023). MGD received funding from the ARC (Grant DE150100456) and the Spanish Ministry for the Economy, Industry and Competitiveness Ramón y Cajal 2017 grant reference RYC-2017-22964. We acknowledge the World Climate Research Programme's Working Group on Coupled Modelling, which is responsible for CMIP, and we thank the climate modelling groups (listed in Table 1 of this paper) for producing and making their model output available.

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

**Table 1** List of CMIP5 models used in this study and their institution.

| Model | Modelling group |
|---|---|
| CanESM2 | Canadian Centre for Climate Modelling and Analysis (CCCMA) |
| CNRM-CM5 | Centre National de Recherches Météorologiques / Centre Européen de Recherche et Formation Avancée en Calcul Scientifique (CNRM-CERFACS) |
| CSIRO-Mk3.6.0 | CSIRO in collaboration with Queensland Climate Change Centre of Excellence (CSIRO-QCCCE) |
| INM-CM4 | Institute for Numerical Mathematics (INM) |
| MPI-ESM-LR | Max Planck Institute for Meteorology (MPI-M) |
| MPI-ESM-MR | Max Planck Institute for Meteorology (MPI-M) |

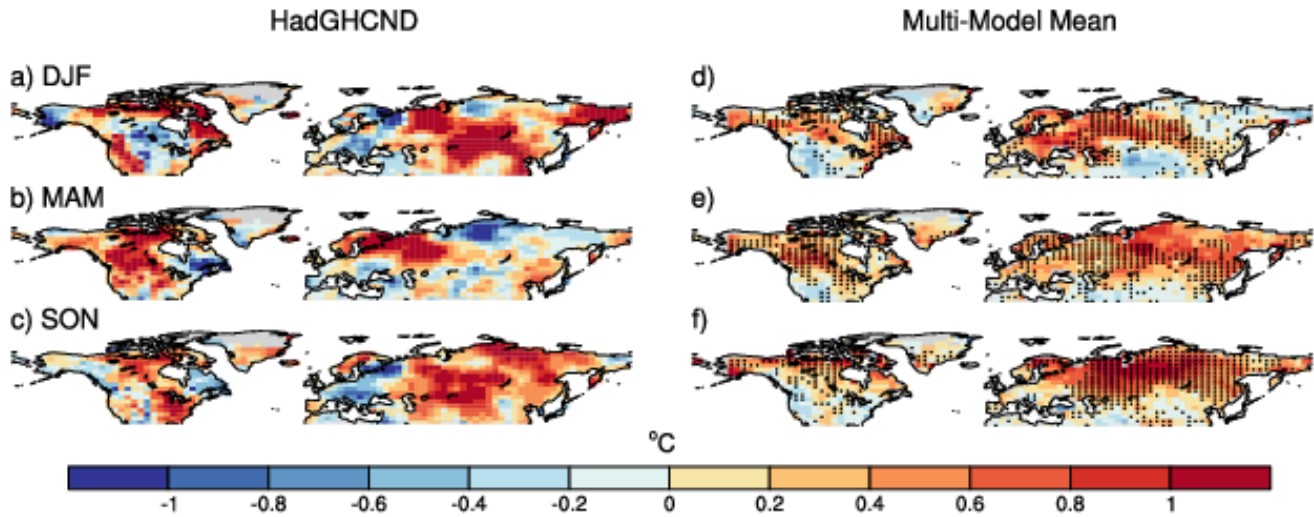

**Figure 1:** Recent excess changes (1982-2014 – 1950-1981) in cold extremes (seasonal minima – seasonal mean) in HadGHCND (a – c) and the six-member CMIP5 multi-model mean (d – f) for boreal winter (a, d), spring (b, e) and autumn (c, f). Grey areas represent areas where data is missing in HadGHCND. Stippling in the multi-model mean represents grid cells where both the multi-model mean agrees in sign with HadGHCND, and where at least five out of six models agree on the sign of excess change.

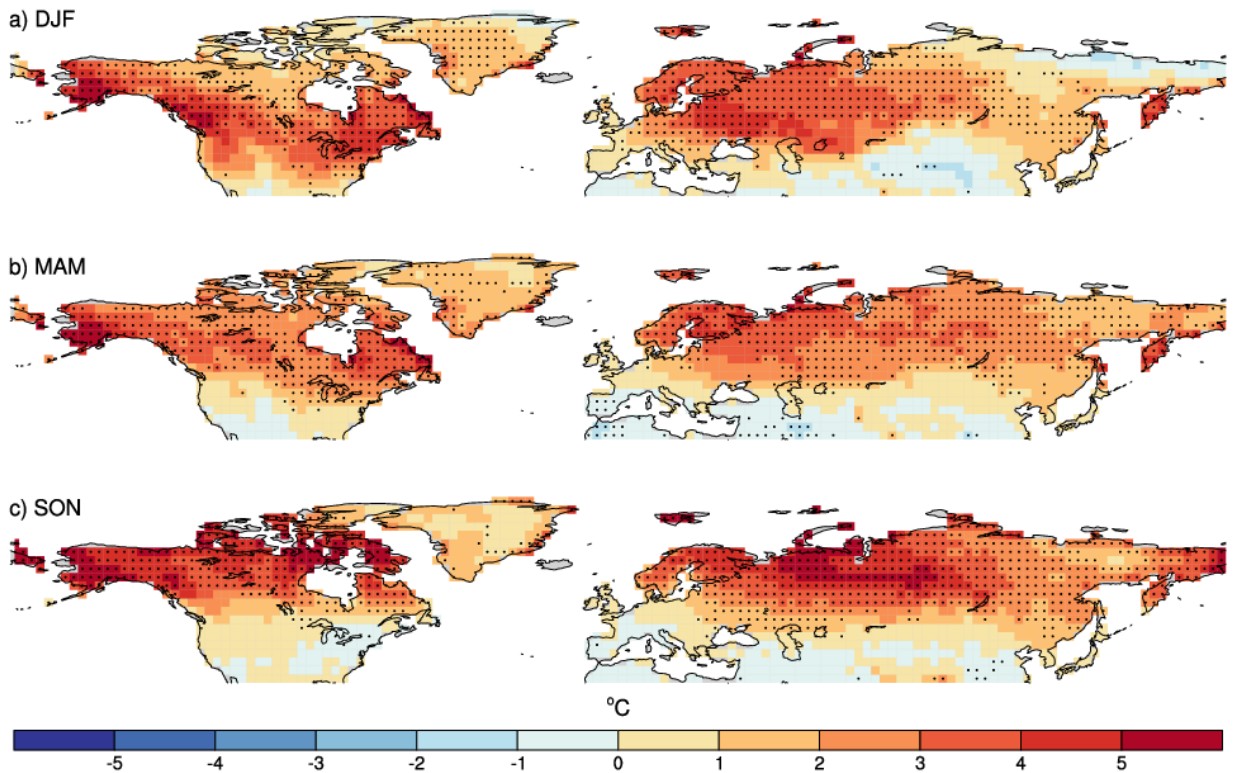

**Figure 2:** Future excess changes (2070-2099 – 1950-1979) in cold extremes (seasonal minima – seasonal mean) in the six-member CMIP5 multi-model mean for boreal winter (a), spring (b) and autumn (c). Stippling indicates grid cells that are both significant at the 5% level as assessed by a KS-test, and where at least five out of six models agree on the sign of excess change.

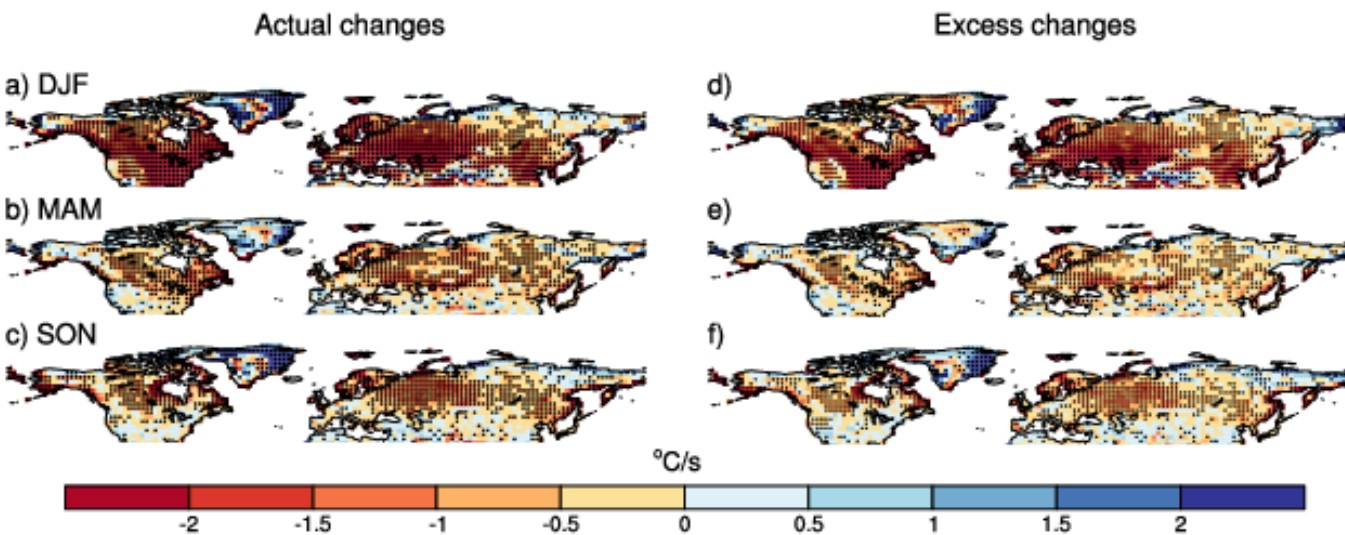

**Figure 3:** Projected future changes (2070-2099 – 1950-1979) in actual (a – c) and excess (d – f) negative temperature advection in °C/s in the six-member CMIP5 multi-model mean for boreal winter (a, d), spring (b, e) and autumn (c, f). Changes are calculated using the average negative temperature advection for the three days prior to the day the seasonal minimum occurs, with negative values indicating reductions in cold air advection, and positive values indicating increases.

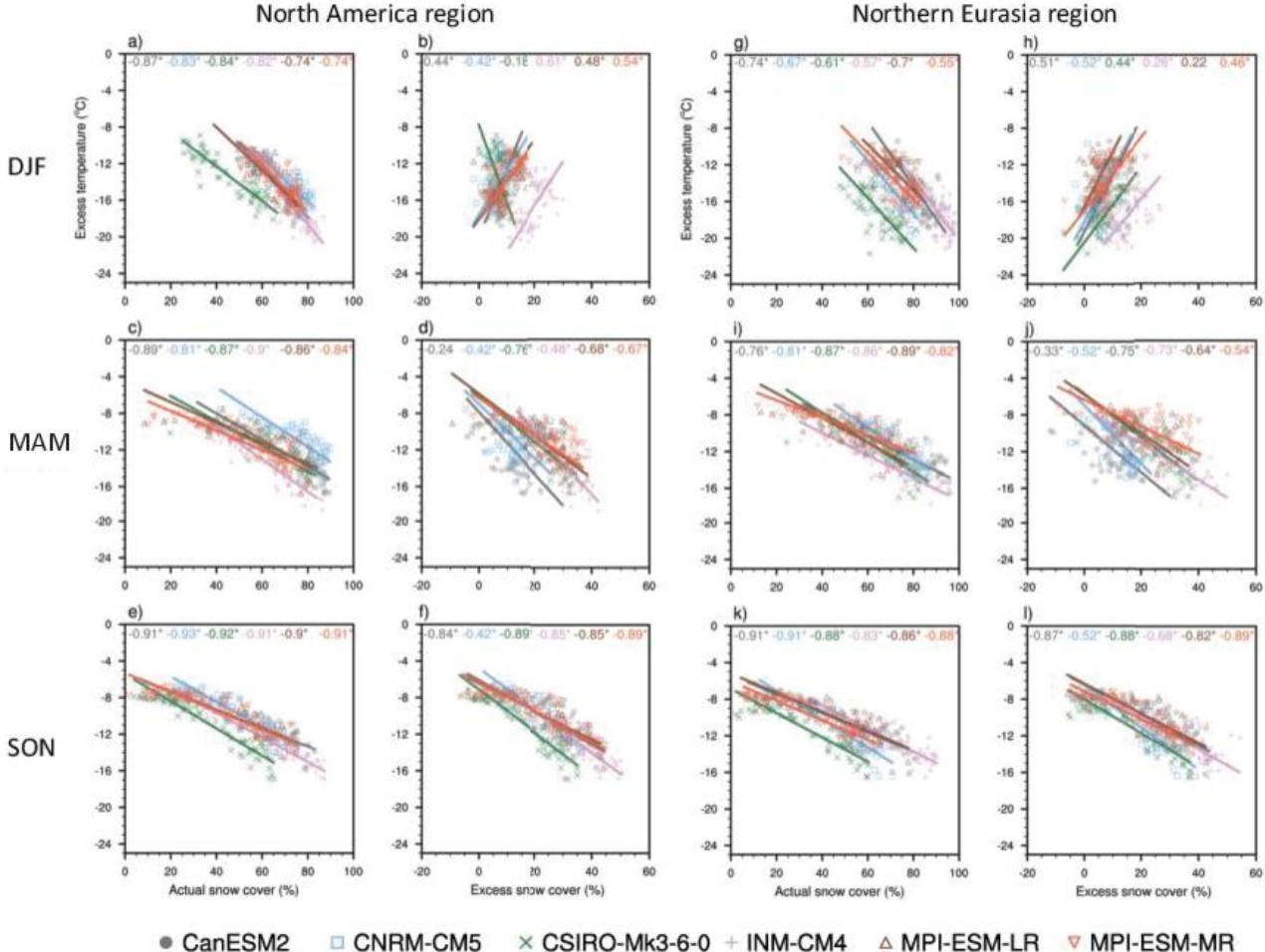

**Figure 4:** Scatter plots showing annual values of excess temperatures in cold extremes for each season on the y-axis, and annual values in each season of actual snow cover (snow cover values on the day the cold extreme occurs) on the x-axis (1st and 3rd column). The 2nd and 4th column show values of excess snow cover on the x-axis (i.e. snow cover on the day of the extreme – mean seasonal snow cover). Each row represents a different season: boreal winter (DJF) in the top row, spring (MAM) in the middle row, and autumn (SON) in the bottom row. Each point is an area-average of two regions (see Fig. S2): North America (a-f) and northern Eurasia (g-l). The straight lines indicate the regression slope for each model calculated using total least squares regression. Correlation coefficients are shown at the top of each panel, with the different colours indicating the model. * indicates significance at the 5% level.

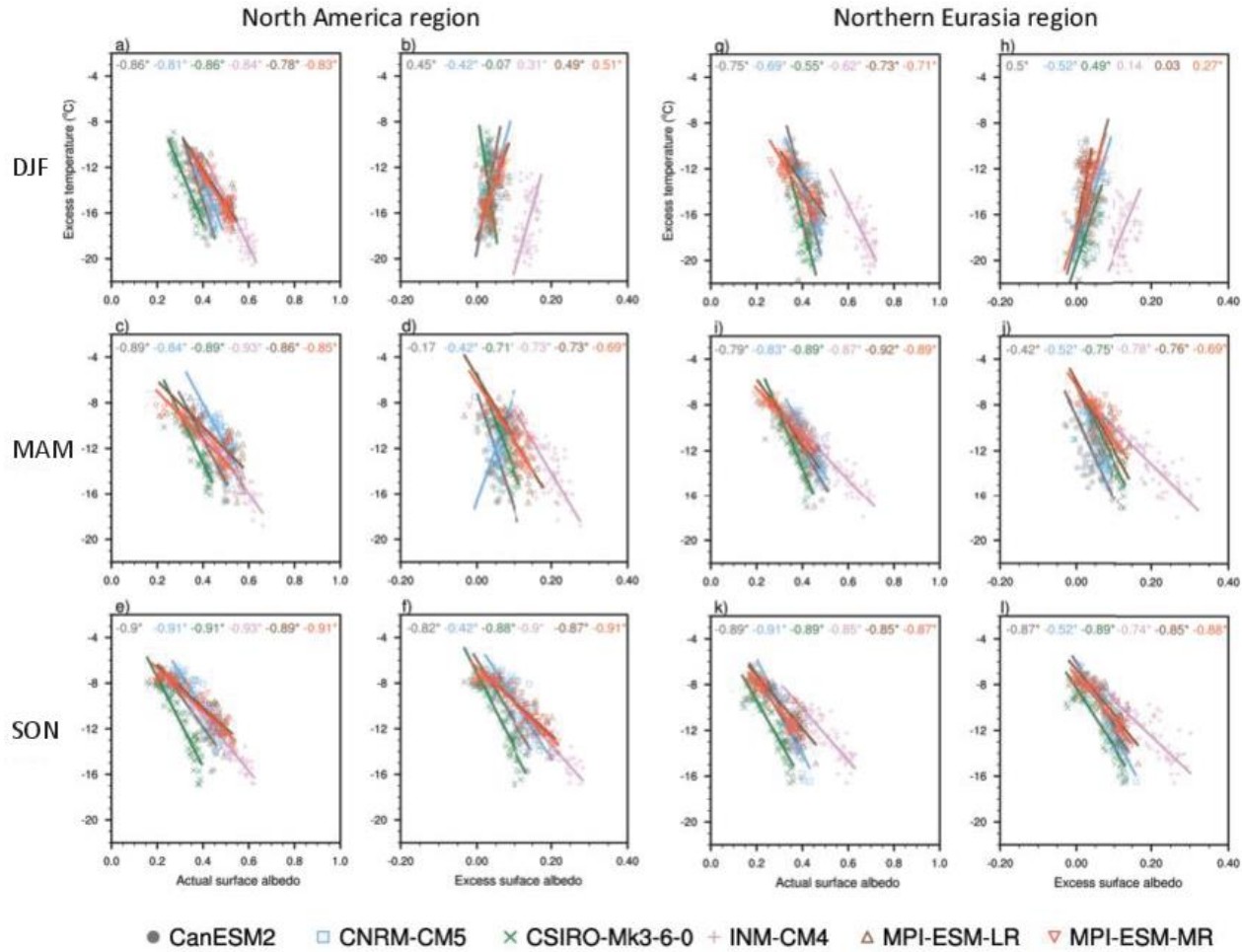

**Figure 5:** As Fig. 4, but for surface albedo.

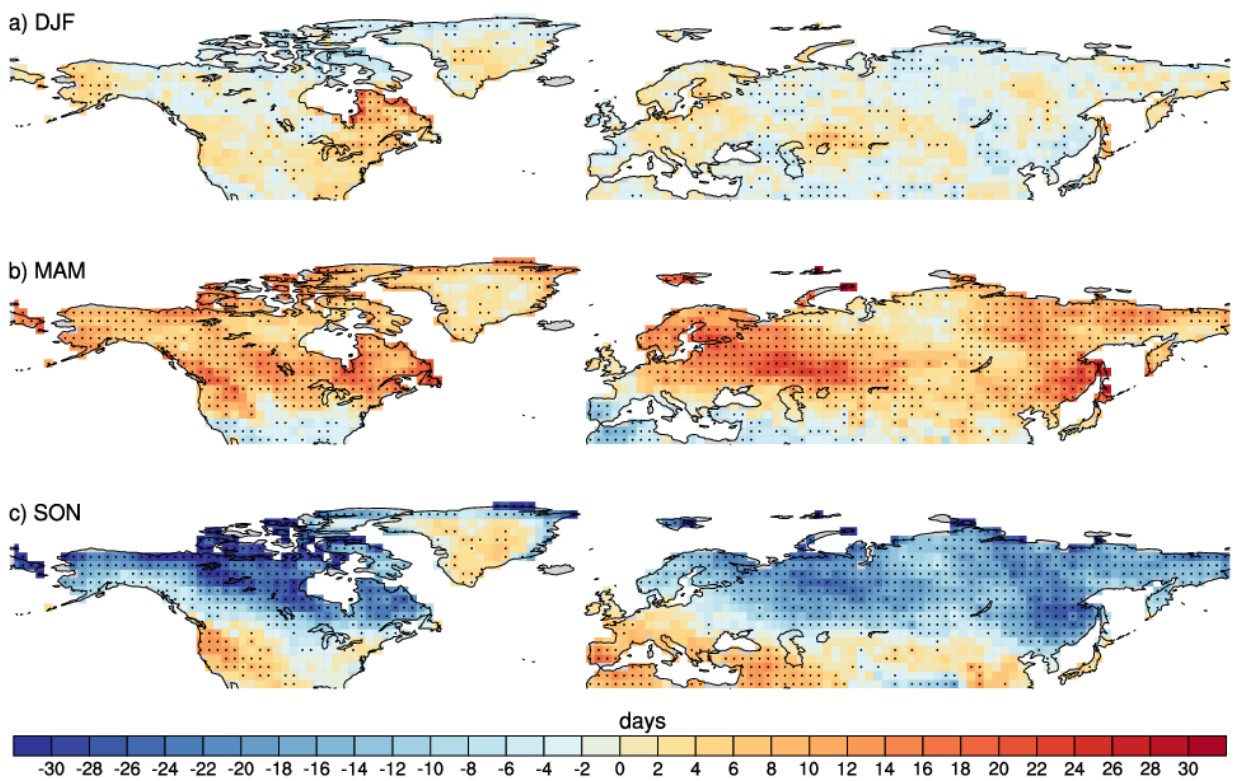

**Figure 6:** Projected changes in the timing of the anomalously coldest day of the season between 2070-2099 and 1950-1979 in the six-member CMIP5 multi-model mean for boreal winter (a), spring (b) and autumn (c). Positive values indicate grid cells where the anomalously coldest day occurs later in the season, while negative values indicate grid cells where the anomalously coldest day occurs earlier in the season. Stippling indicates where at least five out of the six models agree on the sign of change.