# Peer review of "Amplified warming of seasonal cold extremes relative to the mean in"

_Earth System Dynamics, 2019_

## Referee Comment (RC1) · Anonymous Referee #1 · 19 Aug 2019

Review of 'Enhanced warming of seasonal cold extremes relative to the mean in the Northern Hemisphere extratropics' by Mia H. Gross et al.

The submission has the potential to make a significant contribution to the literature, but it is not quite there yet.

For the most part the standard of English in the submission is quite good. However, there are some instances where the expression is awkward or that the meaning is unclear. It would be valuable for the authors to seek the help of a colleague who is proficient in written English when preparing their revision

This manuscript explores the extent to which our cold extremes are getting warmer,

in both absolute terms and relative to the local environmental warming. Investigation makes use observations and also simulations of the future in 6 CMIP5 climate models (RCP8.5 scenario). A valuable part of the investigation is that the authors suggest the physical processes and drivers which lie behind the changes they document.

Before acceptance could be recommended, there are a number of issues which need to be addressed.

Line 1: I think the word 'Enhanced' in the title is redundant (and potentially confusing). (I have a similar issue with this wording at numerous places in the manuscript (e.g., Lines 44-45, . . ..)

Line 17: potential (sp.)

Line 25: It is not clear to me that the paper uses a 'novel' approach. Lines 95, . . . indicate the authors are following earlier studies. Either justify this statement or remove the word here.

Lines 54-60: With relevance to AA and the role of the northerlies in cold extremes it is worth referencing here the recent study of . . . Yuki Kanno, John E. Walsh, Muhammad R. Abdillah, Junpei Yamaguchi and Toshiki Iwasaki, 2019: Indicators and trends of polar cold airmass. Environmental Research Letters, 14, 025006, doi: 10.1088/1748-9326/aaf42b. They show that in the Arctic the loss of extremely cold air is happening at a faster rate than the loss of moderately cold air. Also reference Screen and co-authors, 2018: Polar climate change as manifest in atmospheric circulation. Curr. Clim. Change Reps., 4, 383-395 in this broader context.

Line 63-64: Also include in this citation list the paper of Screen et al., 2014: Amplified mid-latitude planetary waves favour particular regional weather extremes. Nature Clim. Change, 4, 704-709.

Line 88: How can we 'improve' future projections when we don't know the future. Reword this phrase more appropriately.

Line 109-113: I am surprised that natural variability (as revealed in intra-model (or ensemble member) differences) is deemed to be small in this context of cold events. While the use of the first member of the six models is OK, the reader is entitled to some quantitative justification for this statement.

Line 148-150: Please write this equation in more conventional form, and note that 'advection' should have a negative on the right side of the equation. Also, the associated text needs to be expressed better.

Lines 160-163: I can understand a three-day average prior to the day the cold extreme occurs is employed here as it can be seen as representing the cumulative (in time) effect of the relevant processes. However it is not clear to me why it is applied just to Tadv, and not to snow, albedo etc. The influence of these last would also be imagined as relevant in the days leading up to the event (rather than just consider synchronous conditions). Some extra rationalisation/explanation is warranted here.

Line 184-208: This section should be rethought. The pattern correlation coefficients of models with HadGHCND are quite small. At the very least it should be established whether these r values differ significantly from zero (field significance) when appropriate allowance is made for spatial autocorrelation. Need to convince the reader that we are not just looking at noise here. See, for example, LIVEZEY, R.E. & CHEN, W.Y. 1983. Statistical field significance and its determination by Monte Carlo techniques. Monthly Weather Review, 111, 46-59, doi: 10.1175/1520-0493(1983)111<0046:SFSAID>2.0.CO;2. Wang X, Shen SS (1999) Estimation of spatial degrees of freedom of a climate field. J. Climate 12: 1280-1291 doi: 10.1175/1520-0442(1999)012<1280:eosdof>2.0.co;2. Bretherton CS, Widmann M, Dymnikov VP, Wallace JM, Bladé I (1999) The effective number of spatial degrees of freedom of a time-varying field. J. Climate 12: 1990-2009. Also in this section one must be careful of making a posteriori judgements of the geographical locations where the models seem to be in concert and might be of use (in random data such regions can always be found). Any argument for specific regions should be backed up by some physical reasoning. A related issue is the whether these models have proved their worth (or otherwise) in present climate to be trusted to examine cold extremes under future scenarios.

Lines 238-: I think it is confusing to speak of cold air temperature advection (ll. 238, 247-266, 359, 385, . . .) and cold air advection (ll. 265, caption of Fig. 7, . . .). Referring to just 'temperature advection' (its sign, magnitude etc.) makes the argument much simpler. As a more general (and serious?) comment here, I am a little confused by this Section and what Fig. 7 is actually telling us. At line 243 the authors use the word 'actual' for the first time. I first thought this meant, in this case, the changes in the climatological T advection (left panels in Fig. 7). The right panels show 'excess changes' (where 'the difference between changes in the seasonal minima and changes in the seasonal mean is then calculated, hereafter referred to as "excess changes" ' (lines 135-7). My interpretation seemed to be borne out when the authors stated, in connection with the similarity of the left panels and right panels in Fig. 7, that this . . . 'suggests the changes are related to a change in the overall mean state of cold air temperature advection, rather than changes associated with the days directly prior to the day the cold extreme occurs'. However, if this last statement is true one would have expected the 'excess changes' would be close to zero. All this might be just associated with their choice of words. It is important that this issue is addressed and clarified, as the temperature advection argument is central to the paper.

Line 254 (and Fig. 7): I have a little trouble understanding the units here. They are stated as degC, whereas the units of T advection are degC/sec. Please clarify.

Lines 365-380: Some useful discussion pertaining to the complexities is presented here. Thru here worth reminding the reader of the considerable regions on NEGATIVE excesses over parts of Eurasia (Figs. 1 & 4) and the resemblance of these to the 'Warm Artic-Cold Eurasia' pattern. This warrants some extra comment and reference to recent work of Overland and co-authors (2019) Weakened potential vorticity barrier linked to recent winter Arctic sea ice loss and midlatitude cold extremes. J. Climate,

32(14), 4235-4261 and Luo et al. – 2019: The winter midlatitude-Arctic interaction: Effects of North Atlantic SST and high-latitude blocking on Arctic sea ice and Eurasian cooling. Clim. Dyn., 52(5-6), 2981-3004.

Line 418: Would make more sense to join this paragraph up with the one preceding it.

Line 439-443: Much of the analysis and discussion in the paper is focused (appropriately) on the role of AA in inducing these changes in our hemisphere. Notably absent in this exploration is the role that increased moisture load (and hence enhanced downward longwave radiation) plays in AA and its broader consequences. At the very least this should be mentioned and reference made to the investigations of Lee, Feldstein et al., 2017: Revisiting the cause of the 1989-2009 Arctic surface warming using the surface energy budget: Downward infrared radiation dominates the surface fluxes. Geophys. Res. Lett., 44, 10,654–10,661 AND Luo and co-authors (2017) Atmospheric circulation patterns which promote winter Arctic sea ice decline. Env. Res. Lett. 12, 054017, doi: 10.1088/1748-9326/aa69d0.

Lines 444-446: Important to make clear here that the '1.7 days' referred to here was for the period 1954 to 2007. Also, the Stine paper is now quite old. Please to update this by citing new techniques (and measures of uncertainty). E.g., Qimin Deng, and Zuntao Fu, 2019: Comparison of methods for extracting annual cycle with changing amplitude in climate series. Climate Dynamics, 52, 5059-5070, doi: 10.1007/s00382-018-4432-8. Qimin Deng, Da Nian and Z. Fu, 2018: The impact of inter-annual variability of annual cycle on long-term persistence of surface air temperature in long historical records. Climate Dynamics, 50, 1091-1100, doi: 10.1007/s00382-017-3662-5.

Lines 449-50: Reinforce and complement this comment on the shifts associated with the NAM by pointing to the paper of Luo, Dai, et al., 2017: Winter Eurasian cooling linked with the Atlantic Multidecadal Oscillation. Env. Res. Lett., 12, 125002, doi: 10.1088/1748-9326/aa8de8.

Lines 497-500: Please making minor corrections to author list ... Chapin, F. S., III, M.

Sturm, M. C. Serreze, J. P. McFadden, J. R. Key, A. H. Lloyd, A. D. McGuire, T. S. Rupp, A. H. Lynch, J. P. Schimel, J. Beringer, W. L. Chapman, H. E. Epstein, E. S. Euskirchen, L. D. Hinzman, G. Jia, C.-L. Ping, K. D. Tape, C. D. C. Thompson, D. A. Walker and J. M. Welker, 2005: Role of land-surface changes in Arctic summer warming. Science, 310, 657-660, doi: 10.1126/science.1117368.
* * *

---

## Referee Comment (RC2) · Anonymous Referee #2 · 21 Aug 2019

Review of manuscript # ESD-2019-36 entitled "Enhanced warming of seasonal cold extremes relative to the mean in the Northern Hemisphere extratropics".

Gross et al. evaluate historical and projected changes in cold extremes across the Northern Hemisphere extratropics in a subset of CMIP5 models. The authors find that cold extremes are expected to warm substantially more than the seasonal mean. They attribute this difference to changes in advection during winter, and reductions in snow cover during fall and spring. The manuscript is fairly well written, but there are some instances where the text is repetitive or confusing. It features interesting analysis that is relevant to ESD and could be a useful contribution after some issues are addressed.

[Figure]

Major Comments:

(1) Section 3.1: Given the weak pattern correlations I recommend changing the way in which these results are discussed. The authors should lead by describing the observed pattern and quantifying the mean across some given regions (e.g. hemispheric or continental). Second, they can discuss how well the models capture this on average (again quantifying results, perhaps with an ensemble mean value), then it makes sense to comment on the regional differences and the pattern correlations. For example, how much more have cold extremes warmed than the mean in observations and models across either the NH extratropics or EU vs NA (using regions in Fig. S1)?

(2) Baseline used for calculating future changes: Wouldn't it make more sense to compare 1982-2014 with 2070-2099 to evaluate future changes rather than using 1950-1981? The current definition used to calculate "future changes" combines both historical and 21st century changes in one. I recommend changing the baseline to the later historical period.

(3) Presentation of results: There are a number of figures which can be improved or combined to better convey the authors main points. Figures 1-3 could be combined into one plot that shows the excess changes from HadGHCND and the CMIP5 mean for each season (as columns). A polar stereographic projection might be useful for this purpose. Individual model results could then be shown in the supplemental material or similarly as figure 2 if there are models of particular interest. If the authors agree that the CMIP5 mean is useful, I would also add it to Figs 4-6. Figs 7,10: Using the CMIP5 mean (or similar) would also be more informative than assessing the changes in an individual randomly selected model (CanESM2).

Specific Comments:

L14: Remove "The consequences of".

L19: Remove "months".

[Figure]

L45-46: Remove second use of "in these regions".

L54: Change "depending on the region and season" to "both regionally and seasonally".

L63: Remove "which are warming faster than the warm days" – this has already been established above.

L74-76: Surface albedo feedback stemming from snow cover is strongest in spring not winter (see Fig 1c from Qu and Hall 2014). Furthermore, the timing does not only depend on large snow cover extent, but also insolation (which is very low in high snow cover months like December and January).

L85: Reword the first part of this sentence.

L86: Change "capture" to "represent".

L87-88: This is a very broad statement. Either remove or be more specific about what is meant and explain how so. It is also unclear what is being "influenced".

L143: This is repetitive - remove "daily data for".

L169-170: Remove "one by showing" and "the other by showing".

L172: Remove "due to the number of figures".

L176: Change "that have" to "with".

L177: Define the regions with the latitude/longitude here.

L186: Confusing wording. Change "Excess changes between recent decades" to "Historical excess changes".

L197: "Springtime show" – reword.

L210: Remove "between future and past decades". This should be implied by the use of projected.

Figures 4-6, 10: all have colorbars with an unnecessary level of label precision. Increase the tick mark stride.

L214-215: Awkward wording. Change to something like this: "Cold extremes are projected to warm significantly more than mean temperatures across much of the NH extratropics."

L215: Quantify these results in some way.

L243: Define what is meant by the "actual" change. I assume this is just the change in that variable, but the current wording is awkward and difficult to interpret.

L271: Change "high snow fall" to "seasonal snow cover".

Figure 8: Fix cut-off MAM label, increase resolution of figure. The caption says this is the change in snow cover on the day of the extreme versus the mean annual snow cover, but in the methods, it says that the excess change is in reference to the change in the seasonal mean of a variable. Which is correct?

L300-301: Might be useful to state that the lack of snow cover changes in the coldest climates (e.g. Siberia) is due to the trade-off between increasing temperatures shortening the snow season and increased moisture holding capacity leading to greater snowfall. (e.g., Krasting et al. 2013 JCLIM, Mankin and Diffenbaugh 2015 Clim Dyn).

L311: Remove "which is calculated using shortwave radiation fluxes" as this is already stated elsewhere.

L3334: But it is the DJF cold extremes that are of the greatest importance, right? Why not include this as well to be consistent with the rest of the paper?

L334-345: This point is slightly counterintuitive so I recommend adding more information to help the reader. Plot the historical DOY when cold extremes occur to add context to Fig 10. It might also be worth promoting some of the discussion material on this matter to help support this conclusion. Also, I am surprised that this timing would

not be somewhat sensitive to internal variability.

L365-375: A new paper by Blackport et al. in highly relevant to this discussion (https://www.nature.com/articles/s41558-019-0551-4).

L408-409: Fix this comment on surface albedo feedback as recommended above.

L411-413: Reword this, removing "as simply the ratio between absorbed and reflected shortwave radiation".

L418: Promote this material to the paragraph above.

L467: A number of studies have looked at changes in observed versus simulated snow cover (see Brutel-Vuilmet et al., 2013; Mudryk et al 2017).

References listed in this review:

Brutel-Vuilmet, C., M. Ménégoz, and G. Krinner, 2013: An analysis of present and future seasonal Northern Hemisphere land snow cover simulated by CMIP5 coupled climate models. Cryosph., 7, 67–80, doi:10.5194/tc-7-67-2013. http://www.the-cryosphere.net/7/67/2013/.

Krasting, J. P., A. J. Broccoli, K. W. Dixon, and J. R. Lanzante, 2013: Future changes in northern hemisphere snowfall. J. Clim., 26, 7813–7828, doi:10.1175/JCLI-D-12-00832.1.

Mankin, J. S., and N. S. Diffenbaugh, 2015: Influence of temperature and precipitation variability on near-term snow trends. Clim. Dyn., 45, 1099–1116, doi:10.1007/s00382-014-2357-4.

Mudryk, L. R., P. J. Kushner, C. Derksen, and C. Thackeray, 2017: Snow cover response to temperature in observational and climate model ensembles. Geophys. Res. Lett., 44, doi:10.1002/2016GL071789.

---

## Author Comment (AC1) · 17 Oct 2019

\*Note RC=reviewer comment; AR=author response

RC: Review of 'Enhanced warming of seasonal cold extremes relative to the mean in the Northern Hemisphere extratropics' by Mia H. Gross et al. The submission has the potential to make a significant contribution to the literature, but it is not quite there yet. For the most part the standard of English in the submission is quite good. However, there are some instances where the expression is awkward or that the meaning is unclear. It would be valuable for the authors to seek the help of a colleague who is proficient in written English when preparing their revision. This manuscript explores

the extent to which our cold extremes are getting warmer, in both absolute terms and relative to the local environmental warming. Investigation makes use observations and also simulations of the future in 6 CMIP5 climate models (RCP8.5 scenario). A valuable part of the investigation is that the authors suggest the physical processes and drivers which lie behind the changes they document. Before acceptance could be recommended, there are a number of issues which need to be addressed.

AR: Many thanks to the reviewer for their very helpful comments and suggestions. We have addressed each comment and will be changing the manuscript accordingly. Some comments require substantial changes to the text and figures. We believe the changes strengthen the manuscript and the revised manuscript will make a significant contribution to the literature.

RC: Line 1: I think the word 'Enhanced' in the title is redundant (and potentially confusing). (I have a similar issue with this wording at numerous places in the manuscript (e.g., Lines 44-45, . . ..)

AR: Line 1: We can see how this word may not be the most appropriate word to use and have decided to replace 'enhanced' with 'amplified', both in the title and in several instances throughout the text (e.g. lines 44-45; line 51; line 66).

RC: Line 17: potential (sp.)

AR: Line 17: This will be corrected.

RC: Line 25: It is not clear to me that the paper uses a 'novel' approach. Lines 95, . . . indicate the authors are following earlier studies. Either justify this statement or remove the word here.

AR: Line 25: It is true that a similar approach has previously been used (e.g. Donat et al. 2017) in examining processes on the day in which the extreme occurs, however it has not been done for cold extremes as is done in this paper. Nevertheless, we will change the wording here to read "This study examines the day in which the extreme

**ESDD**
occurs...".

RC: Lines 54-60: With relevance to AA and the role of the northerlies in cold extremes it is worth referencing here the recent study of . . . Yuki Kanno, John E. Walsh, Muhammad R. Abdillah, Junpei Yamaguchi and Toshiki Iwasaki, 2019: Indicators and trends of polar cold airmass. Environmental Research Letters, 14, 025006, doi: 10.1088/1748-9326/aaf42b. They show that in the Arctic the loss of extremely cold air is happening at a faster rate than the loss of moderately cold air. Also reference Screen and co-authors, 2018: Polar climate change as manifest in atmospheric circulation. Curr. Clim. Change Reps., 4, 383-395 in this broader context.

AR: Lines 54-60: Thanks to the reviewer for the suggested references. We agree these references are certainly appropriate to reference the role of northerlies in cold extremes and Arctic amplification and adding these references helps to strengthen the argument put forward here.

RC: Line 63-64: Also include in this citation list the paper of Screen et al., 2014: Amplified mid-latitude planetary waves favour particular regional weather extremes. Nature Clim. Change, 4, 704-709.

AR: Line 63-64: Thanks to the reviewer again for the suggested reference which is indeed relevant to the text here.

RC: Line 88: How can we 'improve' future projections when we don't know the future. Reword this phrase more appropriately.

AR: Line 88: Yes, this is a fair point. We will reword this to read "...may help to increase confidence in future projections of warming.".

RC: Line 109-113: I am surprised that natural variability (as revealed in intra-model (or ensemble member) differences) is deemed to be small in this context of cold events. While the use of the first member of the six models is OK, the reader is entitled to some quantitative justification for this statement.
AR: Line 109-113: We will include a figure of the intra-model differences in supplementary material for reference. In the figure, the intra-model differences are small, suggesting that natural variability is relatively small for cold extremes relative to mean temperature changes.

RC: Line 148-150: Please write this equation in more conventional form, and note that 'advection' should have a negative on the right side of the equation. Also, the associated text needs to be expressed better.

AR: Line 148-150: The equation for temperature advection will be amended as  $\partial T/\partial t$ = -(u ( $\partial T/\partial x$ )+v ( $\partial T/\partial y$ )) and the associated text will similarly be amended to describe this better i.e. "where  $\partial T/\partial t$  is the horizontal temperature advection in °C/s, u and v are the zonal and meridional wind components (uas and vas, respectively), and  $\partial T/\partial x$  and  $\partial T/\partial y$  are the temperature gradients in the zonal and meridional direction."

RC: Lines 160-163: I can understand a three-day average prior to the day the cold extreme occurs is employed here as it can be seen as representing the cumulative (in time) effect of the relevant processes. However it is not clear to me why it is applied just to Tadv, and not to snow, albedo etc. The influence of these last would also be imagined as relevant in the days leading up to the event (rather than just consider synchronous conditions). Some extra rationalisation/explanation is warranted here.

AR: Line 160-163: We agree with the reviewer that it might be odd that we only applied the 3-day averaging to Tadv. We did look at the 3-day average for snow and albedo and found that there was no real difference in the result (compared to looking at the variable on the day the extreme occurs). In addition, our reasoning was that the radiative effects would be more important on the day in question, although we acknowledge that there could be some cumulative effects in certain weather conditions. However, given that the results are so similar between the 3-day averaging prior to the day and results on the day of the extreme itself, we prefer to keep the results as they are in the paper, but will add further explanation of why the Tadv and other variables are treated slightly
differently (at line 164): "Days leading up to the cold event were also examined for snow cover and albedo, but results showed no clear difference compared to using values on the exact day of the event. In addition, the radiative effects influencing surface temperature would be more relevant on the day the extreme occurs. For these reasons, we show results of snow cover and albedo on the day the extreme occurs."

RC: Line 184-208: This section should be rethought. The pattern correlation coefficients of models with HadGHCND are quite small. At the very least it should be established whether these r values differ significantly from zero (field significance) when appropriate allowance is made for spatial autocorrelation. Need to convince the reader that we are not just looking at noise here. See, for example, LIVEZEY, R.E. & CHEN, W.Y. 1983. Statistical field significance and its determination by Monte Carlo techniques. Monthly Weather Review, 111, 46-59, doi: 10.1175/1520- 0493(1983)1112.0.CO;2. Wang X, Shen SS (1999) Estimation of spatial degrees of freedom of a climate field. J. Climate 12: 1280-1291 doi: 10.1175/1520-0442(1999)0122.0.co;2. Bretherton CS, Widmann M, Dymnikov VP, Wallace JM, Bladé I (1999) The effective number of spatial degrees of freedom of a time-varying field. J. Climate 12: 1990-2009. Also in this section one must be careful of making a posteriori judgements of the geographical locations where the models seem to be in concert and might be of use (in random data such regions can always be found). Any argument for specific regions should be backed up by some physical reasoning. A related issue is the whether these models have proved their worth (or otherwise) in present climate to be trusted to examine cold extremes under future scenarios.

AR: Lines 184-208 (Section 3.1): We agree this section needs to be rethought. We have since decided that the pattern correlation coefficients shown in the figures are not an adequate method of showing the similarities/differences between the individual models and HadGHCND, because in these figures, we want to know regions where models agree/disagree, rather than a regional average that the pattern correlations infer. We have decided to show the multi-model mean instead of individual figures with
conditional stippling added to show agreement between models as well as with observations. This also helps to keep figures more concise and reduces the overall number of figures in the paper (though we will include individual model results in supplementary material). In the amended figure 1, agreement/disagreement is much clearer allowing a more logical flow of this section into the next, discussing future changes. This section will be re-written to first discuss observational results, then the regional similarities/differences between the models and observations. This then leads logically into the next section, future projections, where we discuss those regions where there is model agreement (similarly now showing the multi-model mean rather than individual models).

RC: Lines 238-: I think it is confusing to speak of cold air temperature advection (II. 238, 247-266, 359, 385, . . .) and cold air advection (II. 265, caption of Fig. 7, . . .). Referring to just 'temperature advection' (its sign, magnitude etc.) makes the argument much simpler. As a more general (and serious?) comment here, I am a little confused by this Section and what Fig. 7 is actually telling us. At line 243 the authors use the word 'actual' for the first time. I first thought this meant, in this case, the changes in the climatological T advection (left panels in Fig. 7). The right panels show 'excess changes' (where 'the difference between changes in the seasonal minima and changes in the seasonal mean is then calculated, hereafter referred to as "excess changes" (lines 135-7). My interpretation seemed to be borne out when the authors stated, in connection with the similarity of the left panels and right panels in Fig. 7, that this . .

. 'suggests the changes are related to a change in the overall mean state of cold air temperature advection, rather than changes associated with the days directly prior to the day the cold extreme occurs'. However, if this last statement is true one would have expected the 'excess changes' would be close to zero. All this might be just associated with their choice of words. It is important that this issue is addressed and clarified, as the temperature advection argument is central to the paper.

AR: Lines 238-267: We understand the reviewer's confusion here and have decided to
use 'negative temperature advection' throughout, with a sentence added to the Methods section describing what is meant by this (line 151 – "We refer to cold air temperature advection hereafter as 'negative temperature advection"). The term 'actual' refers to the average of only values of temperature advection on the 3 days prior to the cold extreme, as opposed to the "excess Tadv" (Tadv on extreme days – mean Tadv). We will add a sentence to the Methods section to explain that 'actual' refers to this for simplicity at line 174 – "For simplicity, we use the term 'actual values' to refer to the seasonal mean calculated from values taken on the day the cold extreme occurs." This indeed required clarification, thanks to the reviewer for pointing this out.

RC: Line 254 (and Fig. 7): I have a little trouble understanding the units here. They are stated as degC, whereas the units of T advection are degC/sec. Please clarify.

AR: Line 254, Fig. 7: Yes, the reviewer is correct, the units are degC/sec. We will correct this in text and figures.

RC: Lines 365-380: Some useful discussion pertaining to the complexities is presented here. Thru here worth reminding the reader of the considerable regions on NEGATIVE excesses over parts of Eurasia (Figs. 1 & 4) and the resemblance of these to the 'Warm Artic-Cold Eurasia' pattern. This warrants some extra comment and reference to recent work of Overland and co-authors (2019) Weakened potential vorticity barrier linked to recent winter Arctic sea ice loss and midlatitude cold extremes. J. Climate, 32(14), 4235-4261 and Luo et al. – 2019: The winter midlatitude-Arctic interaction: Effects of North Atlantic SST and high-latitude blocking on Arctic sea ice and Eurasian cooling. Clim. Dyn., 52(5-6), 2981-3004.

AR: Lines 365-380: Thank you to the referee for this very useful suggestion. We agree that this section warrants some extra discussion on the negative excesses in some regions and the 'Warm Arctic-Cold Eurasia' pattern, and in addition, the suggested references are relevant to the current discussion here and will be cited appropriately. Some extra discussion will be added at line 380 as follows: "There are still small re-
gions which show negative excess changes in Eurasia, such as in central-eastern Asia and northern parts of Siberia. This is consistent with the 'warm Arctic, cold Eurasia' pattern and relates to a substantial decline in sea ice concentration in the Barents-Kara seas and high-latitude blocking associated with a positive phase of the North Atlantic Oscillation (Luo and Chen, 2019; Luo et al., 2019)."

RC: Line 418: Would make more sense to join this paragraph up with the one preceding it.

AR: Line 418: Agreed, we will join this paragraph with the one preceding it.

RC: Line 439-443: Much of the analysis and discussion in the paper is focused (appropriately) on the role of AA in inducing these changes in our hemisphere. Notably absent in this exploration is the role that increased moisture load (and hence enhanced downward longwave radiation) plays in AA and its broader consequences. At the very least this should be mentioned and reference made to the investigations of Lee, Feldstein et al., 2017: Revisiting the cause of the 1989-2009 Arctic surface warming using the surface energy budget: Downward infrared radiation dominates the surface fluxes. Geophys. Res. Lett., 44, 10,654–10,661 AND Luo and co-authors (2017) Atmospheric circulation patterns which promote winter Arctic sea ice decline. Env. Res. Lett. 12, 054017, doi: 10.1088/1748-9326/aa69d0.

AR: Line 439-443: We will add a brief discussion here on other potential driving factors such as enhanced downward longwave radiation from increased moisture. Thank you for the suggestion. We did look into each of the surface radiation fluxes as well as net radiation, and found that downward longwave radiation was amplified on the day of the cold extreme, as well as projected increases in the seasonal mean downward longwave radiation. We will add some extra discussion at line 441 as follows: "... for example, increased moisture load and associated enhanced downward longwave radiation have been shown to play an important role in Arctic amplification (Lee et al., 2017; Luo et al., 2017)."
RC: Lines 444-446: Important to make clear here that the '1.7 days' referred to here was for the period 1954 to 2007. Also, the Stine paper is now quite old. Please to update this by citing new techniques (and measures of uncertainty). E.g., Qimin Deng, and Zuntao Fu, 2019: Comparison of methods for extracting annual cycle with changing amplitude in climate series. Climate Dynamics, 52, 5059-5070, doi: 10.1007/s00382-018-4432-8. Qimin Deng, Da Nian and Z. Fu, 2018: The impact of inter-annual variability of annual cycle on long-term persistence of surface air temperature in long historical records. Climate Dynamics, 50, 1091-1100, doi: 10.1007/s00382-017-3662-5.

AR: Lines 444-446: Thanks to the reviewer for pointing this out and for the suggested reference. We will clarify that the reference period for the '1.7 days' referred to in the manuscript is 1954 to 2007. We will update this with the suggested references which are more recent and useful for the discussion here.

RC: Lines 449-50: Reinforce and complement this comment on the shifts associated with the NAM by pointing to the paper of Luo, Dai, et al., 2017: Winter Eurasian cooling linked with the Atlantic Multidecadal Oscillation. Env. Res. Lett., 12, 125002, doi: 10.1088/1748-9326/aa8de8.

AR: Lines 449-450: Thanks again for the suggested reference and NAM information. This is indeed relevant to the discussion here and this will be amended.

RC: Lines 497-500: Please making minor corrections to author list . . . Chapin, F. S., III, M. Sturm, M. C. Serreze, J. P. McFadden, J. R. Key, A. H. Lloyd, A. D. McGuire, T. S. Rupp, A. H. Lynch, J. P. Schimel, J. Beringer, W. L. Chapman, H. E. Epstein, E. S. Euskirchen, L. D. Hinzman, G. Jia, C.-L. Ping, K. D. Tape, C. D. C. Thompson, D. A. Walker and J. M. Welker, 2005: Role of land-surface changes in Arctic summer warming. Science, 310, 657-660, doi: 10.1126/science.1117368.

AR: Lines 497-500: Thanks to the reviewer for pointing this out. We have gone through the author list and have made necessary changes to this reference: Chapin F. S. III,
Sturm, M., Serreze, M. C., McFadden, J. P., Key, J. R., Lloyd, A. H., McGuire, A. D., Rupp, T. S., Lynch, A. H., Schimel, J. P., Beringer, J., Chapman, W. L., Epstein, H. E., Euskirchen, E. S., Hinzman, L. D., Jia, G., Ping, C.-L., Tape, K. D., Thompson, C. D. C., Walker, D. A., and Welker, J. M.: Role of land-surface changes in Arctic summer warming, Science, 310, 657–660, doi: 10.1126/science.1117368, 2005.

---

## Author Comment (AC2) · 17 Oct 2019

\*Note: RC=reviewer comment; AR=author response

RC: Review of manuscript # ESD-2019-36 entitled "Enhanced warming of seasonal cold extremes relative to the mean in the Northern Hemisphere extratropics". Gross et al. evaluate historical and projected changes in cold extremes across the Northern Hemisphere extratropics in a subset of CMIP5 models. The authors find that cold extremes are expected to warm substantially more than the seasonal mean. They attribute this difference to changes in advection during winter, and reductions in snow cover during fall and spring. The manuscript is fairly well written, but there are some

instances where the text is repetitive or confusing. It features interesting analysis that is relevant to ESD and could be a useful contribution after some issues are addressed.

AR: Thank you to the reviewer for their very helpful comments and suggestions. Our responses are outlined point-by-point after the reviewers comment, and we believe once these changes are made, the readability of the manuscript will be improved, discussions are more relevant and added references contribute to strengthening the arguments put forward in the study.

Major Comments: RC: (1) Section 3.1: Given the weak pattern correlations I recommend changing the way in which these results are discussed. The authors should lead by describing the observed pattern and quantifying the mean across some given regions (e.g. hemispheric or continental). Second, they can discuss how well the models capture this on average (again quantifying results, perhaps with an ensemble mean value), then it makes sense to comment on the regional differences and the pattern correlations. For example, how much more have cold extremes warmed than the mean in observations and models across either the NH extratropics or EU vs NA (using regions in Fig. S1)?

AR: Thanks to the reviewer for their helpful suggestions on how to restructure this section. We agree that some substantial changes are needed to the way in which results are presented and how the section is structured. Firstly, we have decided that the pattern correlations are not an appropriate method for showing the similarities and/or differences between the individual models and HadGHCND as this coefficient is an average over the entire study region. Instead, we first take your suggestion and have decided to show the ensemble mean, which shows stippling for where there is both model agreement and agreement with HadGHCND. This allows us to then discuss similarities/differences by region. As per the reviewers suggestion, we first will discuss observational results, then these similarities/differences in the models. This then logically leads to discussing future projections based on those regions where there is model agreement. **ESDD**
RC: (2) Baseline used for calculating future changes: Wouldn't it make more sense to compare 1982-2014 with 2070-2099 to evaluate future changes rather than using 1950-1981? The current definition used to calculate "future changes" combines both historical and 21st century changes in one. I recommend changing the baseline to the later historical period.

AR: Future changes are calculated for the last 30 years of data (2070-2099) relative to the first 30 years of data (1950-1979). We chose to do this to use the full range of data and compare to an earlier period in the historical period, however, we have also made plots of future changes relative to a more recent 30-year historical period, 1975-2005 (which end when the historical runs end in the CMIP5 models). Results using this later period are similar to those using the earlier historical period, though magnitude of positive excess change is generally around 1°C lower for the later period. We reason to use the earlier period as there is a clearer signal-to-noise ratio, but we understand there may be some confusion in the wording 'future changes' which might suggest we consider the changes starting from the present. This is clarified in the text at line 138 which states that "...'future excess changes' refers to excess changes between the mid-20th century and late 21st century."

RC: (3) Presentation of results: There are a number of figures which can be improved or combined to better convey the authors main points. Figures 1-3 could be combined into one plot that shows the excess changes from HadGHCND and the CMIP5 mean for each season (as columns). A polar stereographic projection might be useful for this purpose. Individual model results could then be shown in the supplemental material or similarly as figure 2 if there are models of particular interest. If the authors agree that the CMIP5 mean is useful, I would also add it to Figs 4-6. Figs 7,10: Using the CMIP5 mean (or similar) would also be more informative than assessing the changes in an individual randomly selected model (CanESM2).

AR: Since we have decided to now show the ensemble mean, Figure 1 now combines Figs1-3 (multi-panel showing DJF, MAM and SON for both HadGHCND and the
ensemble mean). As per the reviewer's suggestion, we will now show the individual model results in supplementary material to highlight that the results are not affected by internal variability (in addition to an added supplementary figure which shows an example of multiple ensemble runs showing the similarities within one model). We will also show the ensemble mean for future projections now (which will become Figure 2). Further, as per the reviewers suggestion, we will now show the CMIP5 mean for Tadv, snow cover and albedo (i.e. current Figs 7,10) rather than showing CanESM2.

Specific Comments: RC: L14: Remove "The consequences of".

AR: This will be removed.

RC: L19: Remove "months".

AR: 'months' will be removed.

RC: L45-46: Remove second use of "in these regions".

AR: Thanks to the reviewer for noticing this. This will be removed.

RC: L54: Change "depending on the region and season" to "both regionally and seasonally".

AR: Thank you for this suggestion, the sentence reads much better with this change.

RC: L63: Remove "which are warming faster than the warm days" – this has already been established above.

AR: This will be removed.

RC: L74-76: Surface albedo feedback stemming from snow cover is strongest in spring not winter (see Fig 1c from Qu and Hall 2014). Furthermore, the timing does not only depend on large snow cover extent, but also insolation (which is very low in high snow cover months like December and January).

AR: The text here states that "the effect of snow cover on surface temperature is

**ESDD**
strongest during spring..." (L77). The confusion here probably stems from the sentence before this, which states that 'the surface albedo feedback from snow cover is more likely to influence winter months and early spring...". We will also add a sentence here relating to insolation being low in winter (Dec-Jan) influencing the surface-albedo feedback from snow cover during these months. Thank you for the clarification.

RC: L85: Reword the first part of this sentence.

AR: The first part of this sentence can be re-worded to "Uncertainties related to biases within climate models are often related to ...".

RC: L86: Change "capture" to "represent".

AR: This will be changed.

RC: L87-88: This is a very broad statement. Either remove or be more specific about what is meant and explain how so. It is also unclear what is being "influenced".

AR: This sentence will be amended as follows: "Evaluating the differences between different physics schemes in individual climate model simulations of snow cover, surface albedo and associated physical processes may help to increase the robustness in future projections of warming."

RC: L143: This is repetitive - remove "daily data for".

AR: This will be removed.

RC: L169-170: Remove "one by showing" and "the other by showing".

AR: These will be removed for readability.

- RC: L172: Remove "due to the number of figures".
- AR: This will be removed as it is unnecessary.
- RC: L176: Change "that have" to "with".

**ESDD**
AR: This will be changed to 'with'.

RC: L177: Define the regions with the latitude/longitude here.

AR: Coordinates will be added in parenthesis for both the North America and Eurasia region. Thanks to the reviewer for this suggestion.

RC: L186: Confusing wording. Change "Excess changes between recent decades" to "Historical excess changes".

AR: Thank you for this suggestion, we agree this is confusing and that 'historical excess changes' is clearer and more succinct.

RC: L197: "Springtime show" – reword.

AR: This will be reworded to "During spring".

RC: L210: Remove "between future and past decades". This should be implied by the use of projected.

AR: Agreed. This will be removed.

RC: Figures 4-6, 10: all have colorbars with an unnecessary level of label precision. Increase the tick mark stride.

AR: These figures are being amended to show the multi-model mean now and we will ensure the label precision is appropriate.

RC: L214-215: Awkward wording. Change to something like this: "Cold extremes are projected to warm significantly more than mean temperatures across much of the NH extratropics."

AR: Changing this sentence as per the reviewers suggestion is much better than the current wording, thank you.

RC: L215: Quantify these results in some way.
AR: This section is being re-written as we now show the multi-model mean instead of individual models.

RC: L243: Define what is meant by the "actual" change. I assume this is just the change in that variable, but the current wording is awkward and difficult to interpret.

AR: The term 'actual' refers to the average of only values of temperature advection on the 3 days prior to the cold extreme, as opposed to the "excess Tadv" (Tadv on extreme days – mean Tadv). We will add a sentence to the Methods section to explain that 'actual' refers to this for simplicity at line 174 – "For simplicity, we use the term 'actual changes' to refer to the seasonal mean calculated from values taken on the day the cold extreme occurs, or for the average of days prior to the cold extreme in the case of temperature advection." This indeed required clarification, thanks to the reviewer for pointing this out.

RC: L271: Change "high snow fall" to "seasonal snow cover".

AR: This will be changed.

RC: Figure 8: Fix cut-off MAM label, increase resolution of figure. The caption says this is the change in snow cover on the day of the extreme versus the mean annual snow cover, but in the methods, it says that the excess change is in reference to the change in the seasonal mean of a variable. Which is correct?

AR: Thanks to the reviewer for noticing this. The figure has been fixed and the resolution increased. Clarification is indeed necessary here. 'Annual' should be 'seasonal' here and this will be fixed in the caption.

RC: L300-301: Might be useful to state that the lack of snow cover changes in the coldest climates (e.g. Siberia) is due to the trade-off between increasing temperatures shortening the snow season and increased moisture holding capacity leading to greater snowfall. (e.g., Krasting et al. 2013 JCLIM, Mankin and Diffenbaugh 2015 Clim Dyn).

AR: Thank you for the information, we do think this is useful to add here with the sup-
porting references. We will add a sentence here as follows (after the preceding sentences which describe the lack of snow cover change in these regions: "... The lack of snow cover changes in the coldest climates, such as Siberia, is due to the trade-off between increasing temperatures that shorten the snow season and increased moisture holding capacity leading to greater snowfall in these regions (e.g. Krasting et al., 2013; Mankin and Diffenbaugh, 2015)."

RC: Remove "which is calculated using shortwave radiation fluxes" as this is already stated elsewhere.

AR: This will be removed.

RC: L334: But it is the DJF cold extremes that are of the greatest importance, right? Why not include this as well to be consistent with the rest of the paper?

AR: While it is true that cold extremes in DJF are the strongest, excess changes in SON/MAM are also strong and it is only these shoulder seasons that show this change in timing (in the current Figure 10). We will add a sentence to clarify why we do not show DJF here.

RC: L334-345: This point is slightly counterintuitive so I recommend adding more information to help the reader. Plot the historical DOY when cold extremes occur to add context to Fig 10. It might also be worth promoting some of the discussion material on this matter to help support this conclusion. Also, I am surprised that this timing would not be somewhat sensitive to internal variability.

AR: We think the confusion here (i.e. the counterintuitiveness the reviewer refers to) stems from using anomalies – where Figure 10 shows the change in timing of anomalously cold days that are relative to a mean annual cycle. When using absolute temperature values which are not subject to an annual cycle, such a change in timing is not evident. The change in timing shown in Figure 10 was also tested for other ensemble members, with all showing similar results indicating little internal variability within the
models. Because our main message here relates to the change in timing in the future, we do not feel it will add anything substantial to the study to show the timing during the historical period.

RC: L365-375: A new paper by Blackport et al. is highly relevant to this discussion (https://www.nature.com/articles/s41558-019-0551-4).

AR: This paper is indeed very relevant. We will add a sentence here regarding the greater influence of atmospheric circulation on mid-latitude cold winters, as opposed to the influence of reduced sea ice, as follows "However, atmospheric circulation is argued to play a more substantial role in influencing cold winters compared with Arctic sea ice loss (Blackport et al., 2019)."

RC: L408-409: Fix this comment on surface albedo feedback as recommended above.

AR: L408-409: We will reiterate here that the surface-albedo feedback is strongest in spring and that insolation is low in winter due to high snow content: "A change in surface albedo feedback, as a result of a change in snow cover, is more likely to influence cold days in early spring due to snow accumulation and low insolation during winter months."

RC: L411-413: Reword this, removing "as simply the ratio between absorbed and reflected shortwave radiation". AR: We will remove this part of the sentence which did not add anything to the discussion here.

RC: L418: Promote this material to the paragraph above.

AR: Thank you for the suggestion, we have decided to join this paragraph to the one above.

RC: L467: A number of studies have looked at changes in observed versus simulated snow cover (see Brutel-Vuilmet et al., 2013; Mudryk et al 2017). References listed in this review: Brutel-Vuilmet, C., M. Ménégoz, and G. Krinner, 2013: An analysis of present and future seasonal Northern Hemisphere land snow cover simulated
by CMIP5 coupled climate models. Cryosph., 7, 67–80, doi:10.5194/tc-7-67-2013. http://www.thecryosphere.net/7/67/2013/. Krasting, J. P., A. J. Broccoli, K. W. Dixon, and J. R. Lanzante, 2013: Future changes in northern hemisphere snowfall. J. Clim., 26, 7813–7828, doi:10.1175/JCLI-D-12- 00832.1. Mankin, J. S., and N. S. Diffenbaugh, 2015: Influence of temperature and precipitation variability on near-term snow trends. Clim. Dyn., 45, 1099–1116, doi:10.1007/s00382- 014-2357-4. Mudryk, L. R., P. J. Kushner, C. Derksen, and C. Thackeray, 2017: Snow cover response to temperature in observational and climate model ensembles. Geophys. Res. Lett., 44, doi:10.1002/2016GL071789.

AR: Many thanks for the suggested references for changes in observed snow cover versus simulated snow cover. We will add some additional information in both the introduction and discussion sections supported by these references. This information indeed fills some gaps that were missing in the submitted manuscript. Some context will be added to the discussion at line 426, and the Mudryk et al. 2017 paper is now cited at line 402, as well as in the introduction at line 87. We will also amend line 467 for future work suggestions as "Future work in understanding the physical mechanisms driving cold extremes would benefit in further evaluation of observational data of snow cover and wind against the model simulations used to project future changes".

**ESDD**

---

## Author Response (AR1)

**Author's response to reviewer comments**

**Anonymous Referee #1**

Review of 'Enhanced warming of seasonal cold extremes relative to the mean in the Northern Hemisphere extratropics' by Mia H. Gross et al. The submission has the potential to make a significant contribution to the literature, but it is not quite there yet. For the most part the standard of English in the submission is quite good. However, there are some instances where the expression is awkward or that the meaning is unclear. It would be valuable for the authors to seek the help of a colleague who is proficient in written English when preparing their revision. This manuscript explores the extent to which our cold extremes are getting warmer, in both absolute terms and relative to the local environmental warming. Investigation makes use observations and also simulations of the future in 6 CMIP5 climate models (RCP8.5 scenario). A valuable part of the investigation is that the authors suggest the physical processes and drivers which lie behind the changes they document.

Before acceptance could be recommended, there are a number of issues which need to be addressed.

Many thanks to the reviewer for their very helpful comments and suggestions. We have addressed each comment point-by-point and have changed the manuscript and figures accordingly. Some comments required substantial changes to the text and figures. We believe the changes strengthen the manuscript and that the revised manuscript will make a relevant contribution to the literature. Line numbers indicated in red refer to the revised manuscript.

Line 1: I think the word 'Enhanced' in the title is redundant (and potentially confusing). (I have a similar issue with this wording at numerous places in the manuscript (e.g., Lines 44-45, . . ..)

Line 1: We can see how this word may not be the most appropriate word to use and have replaced 'enhanced' with 'amplified', both in the title and in several instances throughout the text (e.g. lines 19, 44, 51, 65).

Line 17: potential (sp.)

Line 17: This has been corrected to 'potential'.

Line 25: It is not clear to me that the paper uses a 'novel' approach. Lines 95, . . . indicate the authors are following earlier studies. Either justify this statement or remove the word here.

Lines 24-26: It is true that a similar approach has previously been used (e.g. Donat et al. 2017) in examining processes on the day in which the extreme occurs, however it has not been done for cold extremes as is done in this paper. We have changed the sentence here in any case to "The key findings of this study improve our understanding of the environmental conditions that contribute to the accelerated warming of cold extremes relative to mean temperatures.".

Lines 54-60: With relevance to AA and the role of the northerlies in cold extremes it is worth referencing here the recent study of . . . Yuki Kanno, John E. Walsh, Muhammad R. Abdillah, Junpei Yamaguchi and Toshiki Iwasaki, 2019: Indicators and trends of polar cold airmass. Environmental Research Letters, 14, 025006, doi: 10.1088/1748- 9326/aaf42b. They show that in the Arctic the loss of extremely cold air is happening at a faster rate than the loss of moderately cold air. Also reference

Screen and co-authors, 2018: Polar climate change as manifest in atmospheric circulation. Curr. Clim. Change Reps., 4, 383-395 in this broader context.

Lines 57-65: Thanks to the reviewer for the suggested references. These are appropriate to cite the role of northerlies in cold extremes and Arctic amplification. We have now referenced both Kanno et al., 2019 and Screen et al., 2018 appropriately.

Line 63-64: Also include in this citation list the paper of Screen et al., 2014: Amplified mid-latitude planetary waves favour particular regional weather extremes. Nature Clim. Change, 4, 704-709.

Line 63: Thanks to the reviewer again for the suggested reference which is indeed relevant to the text here. We have now cited Screen et al., 2014.

Line 88: How can we 'improve' future projections when we don't know the future. Reword this phrase more appropriately.

Line 89-90: We have reworded this sentence to "Evaluating the differences and similarities between climate model simulations of snow cover, surface albedo and their influences may help to understand sensitivities and increase confidence in future projections of warming."

Line 109-113: I am surprised that natural variability (as revealed in intra-model (or ensemble member) differences) is deemed to be small in this context of cold events. While the use of the first member of the six models is OK, the reader is entitled to some quantitative justification for this statement.

Line 114: We now include an example of intra-model differences as Figure S1, where the intra-model differences are small. This suggests that natural variability is relatively small for cold extremes relative to mean temperature changes.

Line 148-150: Please write this equation in more conventional form, and note that 'advection' should have a negative on the right side of the equation. Also, the associated text needs to be expressed better.

Lines 150-153: The equation for temperature advection has been amended to $\frac{\partial T}{\partial t} = -(u\frac{\partial T}{\partial x} + v\frac{\partial T}{\partial y})$ and the associated text has been amended to describe this better (lines 148-151): "where ∂T/∂t is the horizontal temperature advection in °C/s, u and v are the zonal and meridional wind components (*uas* and *vas*, respectively), and ∂T/∂x and ∂T/∂y are the temperature gradients in the zonal and meridional direction."

Lines 160-163: I can understand a three-day average prior to the day the cold extreme occurs is employed here as it can be seen as representing the cumulative (in time) effect of the relevant processes. However it is not clear to me why it is applied just to Tadv, and not to snow, albedo etc. The influence of these last would also be imagined as relevant in the days leading up to the event (rather than just consider synchronous conditions). Some extra rationalisation/explanation is warranted here.

Lines 162-165: We agree with the reviewer that it might be odd that we only applied the 3-day averaging to Tadv. We did look at the 3-day average for snow and albedo and found that there was no real difference in the result (compared to looking at the variable on the day the extreme occurs). In addition, our reasoning for using a 3-day average was that the radiative effects would be more important on the day in question, although we acknowledge that there could be some cumulative effects in certain weather conditions. However, given that the results are so similar between the 3-

day averaging prior to the day and results on the day of the extreme itself, we prefer to keep the results as they are, but have added further explanation of why the Tadv and other variables are treated slightly differently (lines 164-165): "A three-day average leading up to the day of the cold event was also assessed for snow cover and albedo, but results showed no clear difference compared to using values on the exact day of the event."

Line 184-208: This section should be rethought. The pattern correlation coefficients of models with HadGHCND are quite small. At the very least it should be established whether these r values differ significantly from zero (field significance) when appropriate allowance is made for spatial autocorrelation. Need to convince the reader that we are not just looking at noise here. See, for example, LIVEZEY, R.E. & CHEN, W.Y. 1983. Statistical field significance and its determination by Monte Carlo techniques. Monthly Weather Review, 111, 46-59, doi: 10.1175/1520-0493(1983)1112.0.CO;2. Wang X, Shen SS (1999) Estimation of spatial degrees of freedom of a climate field. J. Climate 12: 1280-1291 doi: 10.1175/1520- 0442(1999)0122.0.co;2. Bretherton CS, Widmann M, Dymnikov VP, Wallace JM, Bladé I (1999) The effective number of spatial degrees of freedom of a time-varying field. J. Climate 12: 1990-2009. Also in this section one must be careful of making a posteriori judgements of the geographical locations where the models seem to be in concert and might be of use (in random data such regions can always be found). Any argument for specific regions should be backed up by some physical reasoning. A related issue is the whether these models have proved their worth (or otherwise) in present climate to be trusted to examine cold extremes under future scenarios.

Section 3.1: We agree this section needed to be rethought. We decided that the pattern correlation coefficients shown in the figures may not have been the best method of showing the similarities/differences between the individual models and HadGHCND, because in these figures, we want to know regions where models agree/disagree, rather than an area-average that the pattern correlations infer. We have re-written this section (as well as the majority of Section 3). We now show the multi-model mean instead of individual models, with conditional stippling added to show agreement between models as well as with observations. This also helps to keep figures more concise and reduces the overall number of figures in the paper (though we now include individual model results in supplementary material). This section now first discusses observations, then the multi-model mean results and regional similarities/differences between the models and observations. This then leads logically into the next section, future projections, where we discuss those regions where there is model agreement.

Lines 238-: I think it is confusing to speak of cold air temperature advection (ll. 238, 247-266, 359, 385, . . .) and cold air advection (ll. 265, caption of Fig. 7, . . .). Referring to just 'temperature advection' (its sign, magnitude etc.) makes the argument much simpler. As a more general (and serious?) comment here, I am a little confused by this Section and what Fig. 7 is actually telling us. At line 243 the authors use the word 'actual' for the first time. I first thought this meant, in this case, the changes in the climatological T advection (left panels in Fig. 7). The right panels show 'excess changes' (where 'the difference between changes in the seasonal minima and changes in the seasonal mean is then calculated, hereafter referred to as "excess changes"' (lines 135-7). My interpretation seemed to be borne out when the authors stated, in connection with the similarity of the left panels and right panels in Fig. 7, that this . . . 'suggests the changes are related to a change in the overall mean state of cold air temperature advection, rather than changes associated with the days directly prior to the day the cold extreme occurs'. However, if this last statement is true one would have expected the 'excess changes' would be close to zero. All this might be just associated

with their choice of words. It is important that this issue is addressed and clarified, as the temperature advection argument is central to the paper.

Section 3.3: We understand the reviewer's confusion here and have decided to use 'negative temperature advection' throughout, with a sentence added to the Methods section describing what is meant by this (lines 152-153– "We refer to advection of cold air temperature hereafter as 'negative temperature advection'"). The term 'actual' refers to the average of only values of temperature advection on the 3 days prior to the cold extreme, as opposed to the "excess Tadv" (Tadv on extreme days – mean Tadv). This indeed requires clarification in the text. We have added a sentence to the Methods section to explain what 'actual' refers to at lines 177-179 – "For simplicity, we use the term 'actual changes' to refer changes in the actual values of the different variables on the days the cold extremes occur (or the three-day average prior to this day for negative temperature advection)."

RC: Line 254 (and Fig. 7): I have a little trouble understanding the units here. They are stated as degC, whereas the units of T advection are degC/sec. Please clarify.

Revised Fig. 3: The reviewer is correct, the units are degC/sec. We have corrected this in the text and figures.

Lines 365-380: Some useful discussion pertaining to the complexities is presented here. Thru here worth reminding the reader of the considerable regions on NEGATIVE excesses over parts of Eurasia (Figs. 1 & 4) and the resemblance of these to the 'Warm Artic-Cold Eurasia' pattern. This warrants some extra comment and reference to recent work of Overland and co-authors (2019) Weakened potential vorticity barrier linked to recent winter Arctic sea ice loss and midlatitude cold extremes. J. Climate, 32(14), 4235-4261 and Luo et al. – 2019: The winter midlatitude-Arctic interaction: Effects of North Atlantic SST and high-latitude blocking on Arctic sea ice and Eurasian cooling. Clim. Dyn., 52(5-6), 2981-3004.

Lines 418-422: Thank you to the referee for this very useful suggestion. We agree that this section warrants some extra discussion on the negative excesses in some regions and the 'Warm Arctic-Cold Eurasia' pattern. Some extra discussion has been added from line 418 as follows: "Though lacking model agreement, small negative excess changes are projected for parts of Eurasia, such as central-eastern Asia and northern parts of Siberia (Fig. 2a). This is also evident in historical excess changes (Fig. 1a). This is consistent with the 'warm Arctic, cold Eurasia' pattern relating to substantial sea ice concentration in the Barents-Kara seas and high-latitude blocking associated with a positive phase of the North Atlantic Oscillation (Luo et al., 2019a; Luo et al., 2019b). A larger model ensemble would be useful to further quantify whether this pattern is robustly projected for excess cold extremes. "

Line 418: Would make more sense to join this paragraph up with the one preceding it.

Line 456: Agreed, we have joined this paragraph with the one preceding it.

Line 439-443: Much of the analysis and discussion in the paper is focused (appropriately) on the role of AA in inducing these changes in our hemisphere. Notably absent in this exploration is the role that increased moisture load (and hence enhanced downward longwave radiation) plays in AA and its broader consequences. At the very least this should be mentioned and reference made to the investigations of Lee, Feldstein et al., 2017: Revisiting the cause of the 1989-2009 Arctic surface

warming using the surface energy budget: Downward infrared radiation dominates the surface fluxes. Geophys. Res. Lett., 44, 10,654–10,661 AND Luo and co-authors (2017) Atmospheric circulation patterns which promote winter Arctic sea ice decline. Env. Res. Lett. 12, 054017, doi: 10.1088/1748-9326/aa69d0.

Line 480-484: We have added a brief discussion here on other potential driving factors such as enhanced downward longwave radiation from increased moisture. Thank you for the suggestion. We did look into each of the surface radiation fluxes as well as net radiation and found that downward longwave radiation was amplified on the day of the cold extreme, as well as projected increases in the seasonal mean downward longwave radiation. A sentence has been added at lines 481-482 as follows: "For example, increased moisture load and associated enhanced downward longwave radiation have been shown to play an important role in Arctic amplification (Lee et al., 2017; Luo et al., 2017)."

Lines 444-446: Important to make clear here that the '1.7 days' referred to here was for the period 1954 to 2007. Also, the Stine paper is now quite old. Please to update this by citing new techniques (and measures of uncertainty). E.g., Qimin Deng, and Zuntao Fu, 2019: Comparison of methods for extracting annual cycle with changing amplitude in climate series. Climate Dynamics, 52, 5059-5070, doi: 10.1007/s00382-018-4432-8. Qimin Deng, Da Nian and Z. Fu, 2018: The impact of inter-annual variability of annual cycle on long-term persistence of surface air temperature in long historical records. Climate Dynamics, 50, 1091-1100, doi: 10.1007/s00382-017-3662-5.

Lines 487-490: Thanks to the reviewer for pointing this out and for the suggested reference. We have clarified in the text that the reference period for the '1.7 days' is 1954 to 2007. We have also updated this section using the suggested references relating to methods used to detect changes in the annual cycle.

Lines 449-50: Reinforce and complement this comment on the shifts associated with the NAM by pointing to the paper of Luo, Dai, et al., 2017: Winter Eurasian cooling linked with the Atlantic Multidecadal Oscillation. Env. Res. Lett., 12, 125002, doi: 10.1088/1748-9326/aa8de8.

Line 494: Thank you for the suggested reference. This is indeed relevant to the discussion here and we now cite Luo et al., 2017 here.

RC: Lines 497-500: Please making minor corrections to author list . . . Chapin, F. S., III, M. Sturm, M. C. Serreze, J. P. McFadden, J. R. Key, A. H. Lloyd, A. D. McGuire, T. S. Rupp, A. H. Lynch, J. P. Schimel, J. Beringer, W. L. Chapman, H. E. Epstein, E. S. Euskirchen, L. D. Hinzman, G. Jia, C.-L. Ping, K. D. Tape, C. D. C. Thompson, D. A. Walker and J. M. Welker, 2005: Role of land-surface changes in Arctic summer warming. Science, 310, 657-660, doi: 10.1126/science.1117368.

Lines 545-548: Thanks to the reviewer for pointing this out. We have gone through the author list and have corrected this reference.

**Anonymous Referee #2**

Review of manuscript # ESD-2019-36 entitled "Enhanced warming of seasonal cold extremes relative to the mean in the Northern Hemisphere extratropics". Gross et al. evaluate historical and projected changes in cold extremes across the Northern Hemisphere extratropics in a subset of CMIP5 models. The authors find that cold extremes are expected to warm substantially more than the seasonal mean. They attribute this difference to changes in advection during winter, and reductions in snow cover during fall and spring. The manuscript is fairly well written, but there are some instances where the text is repetitive or confusing. It features interesting analysis that is relevant to ESD and could be a useful contribution after some issues are addressed.

Thank you to the reviewer for their very helpful comments and suggestions. Our responses are outlined point-by-point in red, with line numbers in red corresponding to the revised manuscript. We have made substantial changes to the manuscript thanks to reviewer comments and feedback. A re-thought and re-written results section has both increased readability and robustness to the study, and additional references thanks to reviewer's suggestions contribute to strengthening the arguments put forward.

**Major Comments:**

(1) Section 3.1: Given the weak pattern correlations I recommend changing the way in which these results are discussed. The authors should lead by describing the observed pattern and quantifying the mean across some given regions (e.g. hemispheric or continental). Second, they can discuss how well the models capture this on average (again quantifying results, perhaps with an ensemble mean value), then it makes sense to comment on the regional differences and the pattern correlations. For example, how much more have cold extremes warmed than the mean in observations and models across either the NH extratropics or EU vs NA (using regions in Fig. S1)?

Thank you to the reviewer for their helpful suggestions on how to restructure this section. We agree that some substantial changes were necessary in how which results are presented and how the section is structured. Firstly, we have decided that the pattern correlations are not an appropriate method for showing the similarities and/or differences between the individual models and HadGHCND as this coefficient reflects patterns over the entire study region. Instead, we first take your suggestion and have decided to show the ensemble mean, where we include stippling for where there is both model agreement and agreement with HadGHCND. This allows us to then discuss similarities/differences by region. As per the reviewer's suggestion, we first discuss observational results, then these similarities/differences in the models. This then logically leads to discussing future projections based on those regions where there is model agreement.

(2) Baseline used for calculating future changes: Wouldn't it make more sense to compare 1982-2014 with 2070-2099 to evaluate future changes rather than using 1950-1981? The current definition used to calculate "future changes" combines both historical and 21st century changes in one. I recommend changing the baseline to the later historical period.

Future changes are calculated for the last 30 years of data (2070-2099) relative to the first 30 years of data (1950-1979). We chose to do this to use the full range of data and compare to an earlier period in the historical period and thereby maximise the signal, however, we have also made plots of future changes relative to a more recent 30-year historical period, 1975-2005 (which end when the historical runs end in the CMIP5 models). Results using this later period are similar to those using the earlier historical period, though magnitude of positive excess change is generally around 1°C lower for the later period. We reason to use the earlier period as there is a clearer signal-to-noise ratio, but

we understand there may be some confusion in the wording 'future changes' which might suggest we consider the changes starting from the present. This is clarified in the text at lines 140-142 which states that "…'future excess changes' refers to excess changes between the mid-20[th] century (1950-1979) and late 21[st] century (2070-2099)."

(3) Presentation of results: There are a number of figures which can be improved or combined to better convey the authors main points. Figures 1-3 could be combined into one plot that shows the excess changes from HadGHCND and the CMIP5 mean for each season (as columns). A polar stereographic projection might be useful for this purpose. Individual model results could then be shown in the supplemental material or similarly as figure 2 if there are models of particular interest. If the authors agree that the CMIP5 mean is useful, I would also add it to Figs 4-6. Figs 7,10: Using the CMIP5 mean (or similar) would also be more informative than assessing the changes in an individual randomly selected model (CanESM2).

Since we have decided to now show the ensemble mean, Fig. 1 now combines Figs. 1-3 (multi-panel showing DJF, MAM and SON for both HadGHCND and the ensemble mean). As per the reviewer's suggestion, we now show the individual model results as supplementary material to highlight that the results are not affected by internal variability (in addition to an added supplementary figure which shows an example of multiple ensemble runs showing the similarities within one model). We also now show the ensemble mean for future projections (Fig. 2), and similarly for Tadv (Fig. 3) and the timing of cold days (Fig. 6), with individual model results included as supplementary material.

**Specific Comments:**

L14: Remove "The consequences of".

L14: This has been removed and the sentence now begins as "Anomalously warm cold extremes can affect…"

L19: Remove "months".

L18: 'months' has been removed and the start of this sentence now reads "During winter, …".

L45-46: Remove second use of "in these regions".

L45: We have removed this.

L54: Change "depending on the region and season" to "both regionally and seasonally".

L54: Thank you for this suggestion, the sentence reads much better with this change

L63: Remove "which are warming faster than the warm days" – this has already been established above.

L63: This has been removed.

L74-76: Surface albedo feedback stemming from snow cover is strongest in spring not winter (see Fig 1c from Qu and Hall 2014). Furthermore, the timing does not only depend on large snow cover extent, but also insolation (which is very low in high snow cover months like December and January).

L77-78: We have amended the text here and added in some further information low insolation during winter. Thanks to the reviewer for the information. The new sentence is as follows: "The surface albedo feedback stemming from snow cover is strongest during spring because insolation is

low during winter months when snow accumulation is at its highest (Qu and Hall, 2014; Diro et al., 2018)."

L85: Reword the first part of this sentence.

L86: The first part of this sentence can be re-worded to "Uncertainties related to biases within climate models are often related to ...".

L86: Change "capture" to "represent".

L87: This has been changed.

L88-90: This is a very broad statement. Either remove or be more specific about what is meant and explain how so. It is also unclear what is being "influenced".

L87-89: This sentence will be amended as follows: "Evaluating the differences and similarities between climate model simulations of snow cover, surface albedo and their influences may help to understand sensitivities and increase confidence in future projections of warming."

L143: This is repetitive - remove "daily data for".

L146: This has been removed.

L169-170: Remove "one by showing" and "the other by showing".

L172: These have been removed.

L172: Remove "due to the number of figures".

L174: This has been removed as it is unnecessary.

L176: Change "that have" to "with".

L180: This has been changed to 'with'

L177: Define the regions with the latitude/longitude here.

L181-182: Coordinates have been added in parenthesis for both the North America and Eurasia region. Thanks to the reviewer for this suggestion.

L186: Confusing wording. Change "Excess changes between recent decades" to "Historical excess changes".

L190: Thank you for this suggestion, we agree this is confusing and that 'historical excess changes' is clearer and more succinct.

L197: "Springtime show" – reword.

L199: This has been reworded to "In spring…".

L210: Remove "between future and past decades". This should be implied by the use of projected.

L231: Agreed. This entire section has been re-written and this type of phrasing has been kept in mind.

Figures 4-6, 10: all have colorbars with an unnecessary level of label precision. Increase the tick mark stride.

Revised Figures 3, 6, S6-S8: These figures have all been revised and the tick mark stride has been changed accordingly. Thank you for the suggestion.

L214-215: Awkward wording. Change to something like this: "Cold extremes are projected to warm significantly more than mean temperatures across much of the NH extratropics."

Section 3.2: This section has now been re-written.

L215: Quantify these results in some way.

Section 3.2: This section is being re-written as we now show the multi-model mean instead of individual models and we include appropriate quantification.

L243: Define what is meant by the "actual" change. I assume this is just the change in that variable, but the current wording is awkward and difficult to interpret.

L177-179: The term 'actual' refers to the average of only values of temperature advection on the 3 days prior to the cold extreme, as opposed to the "excess Tadv" (Tadv on extreme days – mean Tadv). We have added a sentence to the Methods section to explain that 'actual' refers to this for simplicity at line 177 – "For simplicity, we use the term 'actual changes' to refer to the seasonal mean calculated from values taken on the day the cold extreme occurs, or for the average of days prior to the cold extreme in the case of temperature advection." This indeed required clarification, thanks to the reviewer for pointing this out.

L271: Change "high snow fall" to "seasonal snow cover".

L300: This has been changed.

Figure 8: Fix cut-off MAM label, increase resolution of figure. The caption says this is the change in snow cover on the day of the extreme versus the mean annual snow cover, but in the methods, it says that the excess change is in reference to the change in the seasonal mean of a variable. Which is correct?

Revised Fig. 4 and 5: Thanks to the reviewer for noticing this. The figure has been fixed and the resolution increased. 'Annual' should be 'seasonal' here in the caption and we have amended the caption accordingly.

L300-301: Might be useful to state that the lack of snow cover changes in the coldest climates (e.g. Siberia) is due to the trade-off between increasing temperatures shortening the snow season and increased moisture holding capacity leading to greater snowfall. (e.g., Krasting et al. 2013 JCLIM, Mankin and Diffenbaugh 2015 Clim Dyn).

L330-333: Thank you for the information, we do think this is useful to add here with the supporting references. We will add a sentence here as follows (after the preceding sentences which describe the lack of snow cover change in these regions: "… The lack of snow cover changes in the coldest climates, such as Siberia, is due to the trade-off between increasing temperatures that shorten the snow season and increased moisture holding capacity leading to greater snowfall in these regions (e.g. Krasting et al., 2013; Mankin and Diffenbaugh, 2015)."

L311: Remove "which is calculated using shortwave radiation fluxes" as this is already stated elsewhere.

L344: This has been removed.

L334: But it is the DJF cold extremes that are of the greatest importance, right? Why not include this as well to be consistent with the rest of the paper?

Section 3.5/Fig. 6: We agree it makes more sense to include this for consistency. The revised figure (Fig. 6) now includes DJF.

L334-345: This point is slightly counterintuitive so I recommend adding more information to help the reader. Plot the historical DOY when cold extremes occur to add context to Fig 10. It might also be worth promoting some of the discussion material on this matter to help support this conclusion. Also, I am surprised that this timing would not be somewhat sensitive to internal variability.

Section 3.5: This section has largely been re-written. We think the confusion here (i.e. the counterintuitiveness the reviewer refers to) stems from using anomalies – where the original Fig. 10 (revised Fig. 6) shows the change in timing of anomalously cold days that are relative to a mean annual cycle. When using absolute temperature values which are not subject to an annual cycle, such a change in timing is not evident. The change in timing shown in Fig. 6 was also tested for other ensemble members, with all showing similar results indicating little internal variability within the models. Because our main message here relates to the change in timing in the future, we do not feel it will add anything substantial to the study to show the timing during the historical period.

L365-375: A new paper by Blackport et al. is highly relevant to this discussion (https://www.nature.com/articles/s41558-019-0551-4).

L403-404: This paper is indeed very relevant. We will add a sentence here regarding the greater influence of atmospheric circulation on mid-latitude cold winters, as opposed to the influence of reduced sea ice, as follows "However, atmospheric circulation is argued to play a more substantial role in influencing cold winters compared with Arctic sea ice loss (Blackport et al., 2019)."

L408-409: Fix this comment on surface albedo feedback as recommended above.

L449-450: We reiterate here that the surface-albedo feedback is strongest in spring and that insolation is low in winter due to high snow content: "A change in surface albedo feedback, as a result of a change in snow cover, is more likely to influence cold days in early spring, compared to winter, due to snow accumulation and low insolation during winter months."

L411-413: Reword this, removing "as simply the ratio between absorbed and reflected shortwave radiation".

L453: This has been removed.

L418: Promote this material to the paragraph above.

L456 Thank you for the suggestion, we have decided to join this paragraph to the one above.

RC: L467: A number of studies have looked at changes in observed versus simulated snow cover (see Brutel-Vuilmet et al., 2013; Mudryk et al 2017). References listed in this review:

Brutel-Vuilmet, C., M. Ménégoz, and G. Krinner, 2013: An analysis of present and future seasonal Northern Hemisphere land snow cover simulated by CMIP5 coupled climate models. Cryosph., 7, 67–80, doi:10.5194/tc-7-67-2013. http://www.thecryosphere.net/7/67/2013/.

Krasting, J. P., A. J. Broccoli, K. W. Dixon, and J. R. Lanzante, 2013: Future changes in northern hemisphere snowfall. J. Clim., 26, 7813–7828, doi:10.1175/JCLI-D-12- 00832.1.

Mankin, J. S., and N. S. Diffenbaugh, 2015: Influence of temperature and precipitation variability on near-term snow trends. Clim. Dyn., 45, 1099–1116, doi:10.1007/s00382- 014-2357-4.

Mudryk, L. R., P. J. Kushner, C. Derksen, and C. Thackeray, 2017: Snow cover response to temperature in observational and climate model ensembles. Geophys. Res. Lett., 44, doi:10.1002/2016GL071789.

L510: Many thanks for the suggested references for changes in observed snow cover versus simulated snow cover. We have amended this last sentence and we have also added some additional information supported by these references (at L88, L464-466).